

# Sub-seasonal precipitation forecasts using preceding atmospheric intraseasonal oscillation signals in a Bayesian perspective

Yuan LI[1], Zhiyong WU[1], Hai HE[1], Hao YIN[1]

[1]College of Hydrology and Water Resources, Hohai University, Nanjing 210098, China

*Correspondence to*: Zhiyong WU (wzyhhu@gmail.com)

**Abstract**. Accurate and reliable sub-seasonal precipitation forecasts remain challenging. The atmospheric intraseasonal oscillation (ISO), which is one of the leading sources of sub-seasonal predictability, could be potentially used as predictors for sub-seasonal precipitation forecasts.. However, the relationships between ISO

signals and sub-seasonal precipitation are of high uncertainty. In this study, we first define potential predictors by analyzing the relationship between preceding atmospheric ISO signals and precipitation for 17 hydroclimatic regions over China during the boreal summer monsoon season. The Least Absolute Shrinkage and Selection Operator (LASSO) and stepwise regression approaches are used to narrow down the number of potential predictors. A Bayesian hierarchical model is then established to predict sub-seasonal precipitation. The model

performance is evaluated through a leave one-year-out cross-validation strategy for both deterministic and probabilistic forecasts. The results suggest that the statistical model we built in this study could provide skillful deterministic sub-seasonal precipitation forecasts over southeastern and southwestern hydroclimatic regions at a lead time of 20-25 days. However, the deterministic forecast skills are much lower over northeastern China, owing to the underestimation of intraseasonal variability in these regions. The probabilistic forecasts are more promising,

and the results indicate that the Bayesian hierarchical model could provide skillful and reliable sub-seasonal precipitation forecasts for all hydroclimatic regions from 0-day to 25-day leads. Other sources of sub-seasonal predictability would be included in the future to further improve sub-seasonal precipitation forecast skills.



## 1. Introduction

Accurate and reliable sub-seasonal precipitation forecasts can provide vital information for many management

decisions in water resources, agriculture, and disaster mitigations (Vitart et al., 2012; Vitart and Robertson, 2018).

One approach for sub-seasonal precipitation forecasts is to run dynamical models such as Global Climate Models

(GCMs). Projects such as the Subseasonal-to-Seasonal Prediction Project (S2S) and the Subseasonal Experiment

(SubX) have been lunched to provide sub-seasonal precipitation forecasts with lead time up to 60 days from GCMs

(Pegion et al., 2019; Vitart et al., 2017). However, the sub-seasonal precipitation forecasts derived directly from

GCMs are of low accuracy as the physical equations are always simplified and small-scale processes could not be

well represented in the GCMs (De Andrade et al., 2019). Post-processing is always required to improve the

accuracy and reliability of GCM forecasts before it could be used for other applications. Schepen et al. (2018) and

our previous study (Li et al., 2020) used the Bayesian Joint Probability (BJP) method to post-process sub-seasonal

precipitation forecasts over different regions, and the results suggested that the forecast skills and reliability were

improved compared to raw GCM forecasts. Vigaud et al. (2020) proposed a new spatial correction method to

improve sub-monthly precipitation forecasts derived from multimodel ensembles. Nevertheless, the results also

indicated that the accuracy of post-processed sub-seasonal precipitation forecasts were still limited when the lead

time was beyond 10-14 days.

An alternative approach for sub-seasonal precipitation forecasts is to establish statistical models based on the

relationships between precipitation and preceding atmospheric-oceanic indices. Although dynamical models are

dominant for short- to medium-term forecasts, statistical models are still found to be useful especially for long-term

forecasts (Tuel and Eltahir, 2018; Abbot and Marohasy, 2014; Mekanik et al., 2013; Lü et al., 2011; Kirono et al.,

2010). Schepen et al. (2012) suggested that the lagged climate indices were potentially useful for seasonal

precipitation forecasts over Australia. Plenty statistical algorithms, such as multiple linear regression or canonical

correlation analysis, have been developed for seasonal precipitation forecasts based on the assumption that the

seasonal anomalies are caused by slow-varying sea surface temperature, sea ice, snow cover, and other boundary

conditions (Hwang et al., 2001; Barnston and Smith, 1996; Eden et al., 2015). Totz et al. (2017) proposed a new

cluster-based empirical method to predict winter precipitation anomalies over the European and Mediterranean

Regions, which the sea surface temperature, geopotential height, sea level pressure, snow cover extent, and sea

ice concentration were included as predictors. A random forest based statistical model, which the predictors were



identified from the gridded sea surface temperature, was developed to predict central and south Asia seasonal precipitation (Gerlitz et al., 2016).

However, much fewer statistical models have been built and applied for sub-seasonal precipitation forecasts as the sources of sub-seasonal predictability are not yet fully understood at such a time scale. Compared to seasonal precipitation forecasts, the slow-varying boundary forcings may have limited impact on sub-seasonal precipitation as the time scale is too short. The atmospheric intraseasonal oscillation (ISO), which is the dominant mode of sub-seasonal variability, is one of the leading sources of sub-seasonal predictability (Robertson and Vitart, 2018). The

boreal summer intraseasonal oscillation (BSISO) in the tropics, which is also known as Madden-Julian Oscillation (MJO) in winter, is characterized as a slow-moving system with a period of 30-90 days in the tropical atmosphere (Madden and Julian, 1971, 1972; Zhang, 2005; Woolnough, 2019; Wang and Xie, 1997). The circulation anomalies associated with the intraseasonal oscillation (ISO) are identified to have an impact on monsoon activities and heavy rainfall events (Annamalai and Slingo, 2001; Chen et al., 2004). Zhang et al. (2009) found that the rainfall patterns

in Southeast China were transited from being enhanced to being suppressed when the MJO center moved from the Indian Ocean to the Western Pacific Ocean. Jia et al. (2011) suggested that the MJO influenced the rainfall patterns in China mainly by modulating the circulation in the subtropics and mid-high latitudes in winter. This suggests that the ISO signals could be potentially used for predicting sub-seasonal precipitation not only in tropical regions but extra-tropical regions as well.


Several statistical models have been built to predict sub-seasonal precipitation based on the relationships between ISO signals and precipitation. Hsu et al. (2015) established a set of spatial-temporal projection models (STPMs) to predict sub-seasonal precipitation at a lead time of 10-30 days over southern China. Their results suggested that the forecast skills were still promising at a 20-25-day lead time. Zhu and Li (2017) predicted sub-seasonal

precipitation by constructing STPMs over entire China, and independent forecasts of rainfall anomalies during the period of Olympic Games in 2008 and Shanghai World Expo in 2010 suggested that the STPMs were able to reproduce intraseasonal rainfall patterns at a 20-day lead time. However, we should note that the relationships between ISO signals and precipitation are of high uncertainty for different regions at different lead times. Chen and Wang (2021) suggested that different BSISO events would have distinct impact on monsoon systems. To our best

knowledge, the uncertainties of relationships between preceding ISO signals and sub-seasonal precipitation have



not been fully considered in sub-seasonal precipitation forecasts in previous studies.

There are several ways to address the above challenge. Lepore et al. (2017) established an extended logistic regression model to link the relationship between El Niño-Southern Oscillation (ENSO) and convective storm (SCS) activity. Sohrabi et al. (2021) coupled the large-scale climate indices with a stochastic weather generator to provide ensemble streamflow forecasts. Compared to the above-mentioned traditional probabilistic model solutions, the Bayes-theorem based statistical models are more flexible and more efficient for assessing multiple sources of uncertainties. Wang et al. (2009) proposed a multivariate normal distribution based Bayesian joint probability (BJP) approach to predict seasonal streamflow over Australia using antecedent streamflow, ENSO indices, and other climate indicators as predictors. Peng et al. (2014) utilized the same BJP approach to predict seasonal precipitation over China using lagged oceanic-atmospheric indices. Another Bayes-theorem based approach, the Bayesian hierarchical model (BHM), has also been developed in recent years (Gelman and Hill, 2006). The BHMs are always constructed with several model layers. The predictand is assumed to follow distribution with unknown parameters in the first layer, and the parameters are linked with the predictors using linear regression models in the second layer. The regression coefficients are given hyperprior distributions with the BHMs. The utility of BHMs has been demonstrated in modelling spatiotemporal variability of hydrological variables in many studies (Renard, 2011; Reza Najafi and Moradkhani, 2013; Bracken et al., 2016; Lima and Lall, 2010; Lima and Lall, 2009; Devineni et al., 2013). The BHMs are also used for seasonal predictions in many fields. Chen et al. (2014) used the BHM to predict summer rainfall and streamflow over the Huai River basin, while Chu and Zhao (2007) developed a BHM model to predict seasonal tropical cyclone activity over the Central North Pacific. However, the BHMs have not been used to predict sub-seasonal precipitation before. In this study, we follow a similar BHM structure proposed by Devineni et al. (2013) to predict sub-seasonal precipitation.

China is located in east Asia, and is frequently influenced by rainstorm and flood disasters during the boreal summer monsoon. Accurate and reliable sub-seasonal precipitation forecasts can provide valuable information for mitigating the risks from rainstorm and flood disasters. However, the origin of intraseasonal precipitation variability is of high complexity owing to the mixed impact of tropical convection, forcing of Tibetan Plateau, and mid-high latitude systems (Zhu and Li, 2017). In this study, we first define potential predictors by analyzing the correlations between preceding ISO signals and precipitation for each hydroclimatic region. In a second step, smaller groups of robust





predictors are selected and the Bayesian hierarchical model is established to predict sub-seasonal precipitation.
        The model performance for both deterministic and probabilistic forecasts are evaluated through a leave one-year-
        out cross-validation strategy.

        In the following section, the observations, reanalysis dataset, ISO signal extraction, potential predictor selection,
Bayesian hierarchical model calibration and evaluation are introduced. The deterministic and probabilistic forecast
        skills are presented in Sect. 3. Section 4 discusses the forecast skills, possible mechanism, limitations, and future
        work. Key findings are summarized in Sect. 5.

## 2. Data and Methodology

### 2.1 Data

        In this study, China is divided into 17 hydroclimatic regions as suggested by Lang et al. (2014). The southeastern
        hydroclimatic regions are mostly of temperate and warm/hot summer climate without dry season (Cfb/Cfa), while
        the northwestern regions are mostly arid with limited precipitation (Bwk, Bsh, Bsk climate types) (Peel et al., 2007)
        (Figure 1). The observed precipitation is derived from the Multi-Source Weighted-Ensemble Precipitation, version
2 (MSWEP V2) dataset. The MSWEP V2 dataset is of high spatial (0.1°) and temporal (3 hourly) resolution.
        Compared to other gridded datasets, the MSWEP V2 exhibits more realistic spatial patterns, and higher accuracy
        over land (Wu et al., 2018; Beck et al., 2019). The 0.1° gridded precipitation data is area-weighted averaging
        through 17 hydroclimatic regions over China from May to October. After that, the 3-hourly regional precipiation data
        is summarized to pentad data to reduce the noise and improve the predictability.



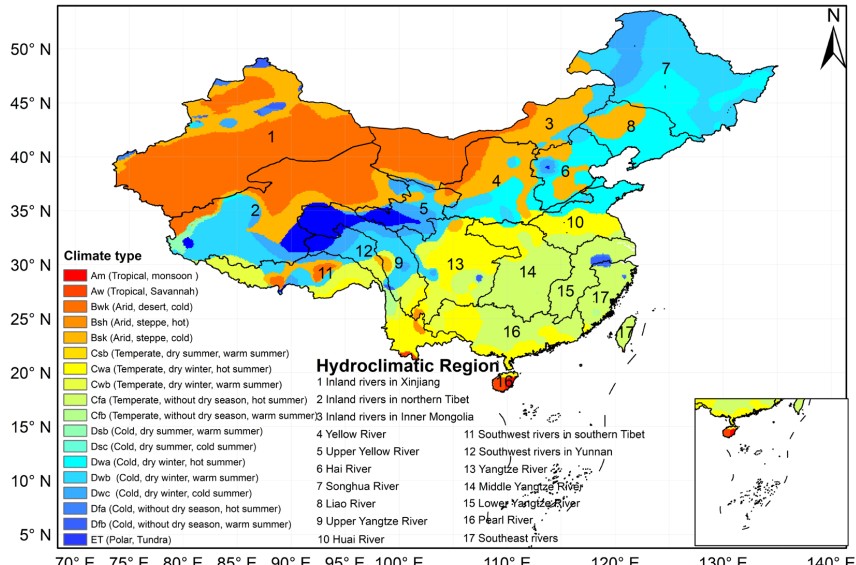

**Figure 1.** 17 hydroclimatic regions over China.

The intraseasonal oscillation is always represented by outgoing longwave radiation (OLR), zonal winds in the upper (200 hPa) and lower (850 hPa) troposphere. Although several indices, including the RMM (Realtime Multivariate MJO) index (Wheeler and Hendon, 2004) and BSISO index (Lee et al., 2013), have been proposed to monitoring the propagation of oscillation, these indices may not cover patterns which might be important for sub-seasonal precipitation in certain regions. To overcome this problem, we analyze the correlation between regional precipitation and preceding global gridded OLR, zonal wind at 850 hPa (U850), zonal wind at 200 hPa (U200) for each grid cell. In addition, the geopotential height at 850 hPa, 500 hPa, and 200 hPa (H850, H500, H200), which have been proved to be capable of reflecting the MJO structure as the zonal wind (Leung and Qian, 2017), are also analyzed.

The OLR-daily Climate Data Record (CDR) used in this study is derived from National Climate Data Center (NCDC) on a 1.0° squared resolution over the globe. The OLR-daily CDR is developed from high resolution infrared radiation sounder instruments, and is valuable to a wide range of applications. A more detailed description of the OLR dataset can be found at https://www.ncdc.noaa.gov/news/new-outgoing-longwave-radiation-climate-data-record. The global gridded daily average U850, U200, H850, H500, H200 data are derived from the ERA5 reanalysis dataset at https://cds.climate.copernicus.eu/. The ERA5 reanalysis dataset is produced using advanced 4D-Var data assimilation scheme, and its horizontal resolution is approximately 30 km with 137 pressure levels in the vertical (Hersbach et al., 2020). It provides hourly record of global atmosphere, land surface and ocean waves from 1950



to present. To focus on large-scale features and increase calculating efficiency, both the OLR-daily CDR dataset and the daily average ERA5 reanalysis dataset are bilinearly interpolated onto 2.5° × 2.5° latitude-longitude resolution. Moreover, we choose to focus on the period of 1979-2016 to be consistent with the temporal coverage of observed precipitation data.

## 2.2 Methodology

### 2.2.1 10-60-day ISO signal extraction

As briefly introduced in previous section, extracting meaningful ISO signals is important for sub-seasonal precipitation forecasts. However, the high-frequency (unpredictable) noises exist for both raw daily precipitation data and raw daily large-scale circulation variables. Band-pass filtering methods, such as the fast Fourier transformation, should be used to isolate intraseasonal low-frequency (10-60-day) signals from raw data (Zhang, 2005). However, traditional band-pass filtering method is impractical for real time applications as future information beyond the current date is needed. In this study, a non-filtering method proposed by Hsu et al. (2015) is used to extract sub-seasonal component with a period between 10 and 60 days for both the ISO variables and precipitation. Compared to traditional ISO signal extraction method, this approach is easy to implement and could be used for real time applications. The climatological annual cycle of raw daily data is first removed by subtracting 90-day low-pass filtered climatological component:

$$x' = x - \bar{x} \qquad (1)$$

where $x$ is the areal-weighted daily precipitation data for each hydroclimatic region or the gridded large-scale circulation variable U850, U200, OLR, H850, H500, or H200, $\bar{x}$ is the corresponding climatological 90-day low-pass filtered component derived by Lanczos filtering method for the period of 1981-2010 (Duchon, 1979). In a second step, lower-frequency signals longer than 60 days are removed by subtracting the last 30-day running mean,

$$x'' = x' - \overline{x'}^{30d} \qquad (2)$$

where $\overline{x'}^{30d}$ is the last 30-day running mean of $x'$.

The higher-frequency signals are then removed by taking a pentad mean,

$$x^* = \overline{x''}^{5d} \qquad (3)$$

The so-derived variables represent the 10-60-day component of ISO signals and precipitation.





### 2.2.2 Defining potential predictors

To identify relevant areas of large-scale circulation fields that could affect 10-60-day precipitation variability, we analyze the correlation between extracted ISO signals in the preceding six pentads and 10-60-day component of precipitation for each hydroclimatic region during the period of 1979-2016. Owing to the data persistence introduced by the filtering method, the effective degree of freedom for each grid cell and each preceding pentad is estimated following Livezey and Chen (1983). The identified significantly correlated preceding ISO signals are then used to establish the Bayesian hierarchical model to produce a 0-25-day lead time precipitation forecasts.

As an example, Figure 2 presents the correlation between preceding U850, U200, OLR, H850, H500, H200 and 10-60-day component of precipitation over Region 1 (Inland Rivers in Xinjiang) at different lead times. At leads 25 to 20 days, the significantly correlated U850 signals are mainly over the Arabian Sea and the Bay of Bengal. The U850 signals are then propagating eastward toward South China Sea and Philippine Sea at the lead of 15 to 10 days. The U850 anomalies then gradually moved eastward and northward toward West Pacific Ocean, Mongolia plateau, Iranian plateau, and Qinghai-Tibet plateau from the lead of 10 to 0 days. The U200 ISO signals are more pronounced compared to U850 signals. The spatial distribution of potential predictive U200 regions is rather concentrated, indicating more robust statistical relationships. The OLR anomalies appear near the Indian Ocean at 20 to 15 day leads. At lead 5 to 0 day, the significantly correlated OLR signals are mainly over the East European Plain and West Siberian Plain. The H850 and H500 high anomalies appear near the Africa at the lead of 20 days to 15 days, and gradually move eastward and northward toward Indian Ocean, Iranian plateau, and Central Asia from the lead of 10 to 0 days. Unlike the H850 and H500 fields that originated over the Africa, the H200 anomaly appears to originate from the Indian Ocean and West Pacific Ocean from leads of 25 to 15 days. At lead 10 to 0 days, the significantly correlated H200 signals are mainly over the East European Plain, West Siberian Plain, and Central Siberian plateau.

To avoid defining too many predictors, which would lead to overfitting, we define potential predictors by averaging U850, U200, OLR, H850, H500, H200 signals in the areas of significant correlations at different lead times. The irregular boundaries of significant correlated areas are identified by the Python package scikit-image (Walt et al., 2014). A total number of about one or two dozen potential predictors are defined for each hydroclimatic region and each preceding pentad.

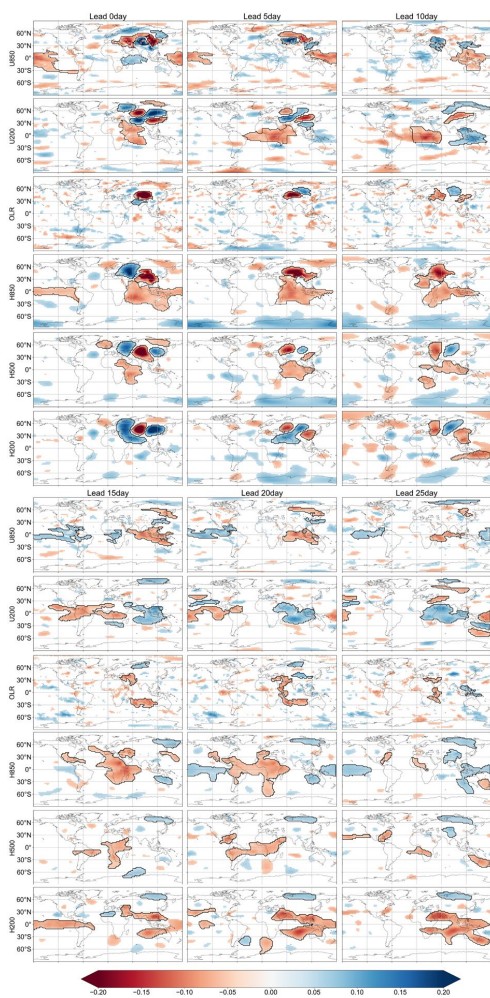

**Figure 2.** Correlation between preceding ISO signals of U850, U200, OLR, H850, H500, H200 and 10-60-day component of precipitation over Region 1 (Inland Rivers in Xinjiang) at different lead times. Correlation coefficients statistically significant at the 5% level are shaded.

### 2.2.3 Statistical modelling

In previous steps, we defined potential predictors by analyzing the relationship between ISO signals and 10-60-day component of precipitation. The so-derived predictors could be used to predict sub-seasonal precipitation amount as well as precipitation indices (Li et al., 2016; Leung and Qian, 2017). Here, the defined predictors are





used to predict pentad mean precipitation amount as it could also be used for hydrological applications. Consider, for example, predicting pentad mean precipitation for the period between 1th May and 5th May, 1979. In this case, ISO signals extracted on 30th April, 25th April, 20th April, 15th April, 10th April, 5th April 1979 are used as predictors to produce precipitation forecasts at different lead times. A leave-one-year-out cross-validation strategy is

implemented for both potential predictor selection, normalization, model building, and verification to avoid any bias in skill (Michaelsen, 1987). For instance, to produce sub-seasonal precipitation forecasts in 1979, the predictors (preceding ISO signals) and predictand (pentad mean precipitation) during the period of 1980-2016 are pooled together for statistical modelling. The forecasts for the year 1979 are then issued by models trained on 1980-2016, and the performance is evaluated against the observations. This cross-validation strategy ensures that the data

used for evaluation is never used for statistical modelling.

**Potential predictor selection**

The number of potential predictors defined previously is still too large for statistical modelling. To narrow down the number of potential predictors, we first use the Least Absolute Shrinkage and Selection Operator (LASSO)

regression to select a smaller subset of robust potential predictors. LASSO is a regularization method that reduces the absolute value of large coefficients. It has been proved to be efficient in selecting predictors and reducing model complexity in many fields (Hammami et al., 2012; Chu et al., 2020). A detailed description of LASSO could be found in (Nardi and Rinaldo, 2011; Mcneish, 2015). After that, we utilize a combined forward and backward variable selection process, which is also known as the stepwise regression, to further select the most informative predictors

for each hydroclimatic region and each preceding pentad.

**Bayesian hierarchical model framework**

Before establishing the Bayesian hierarchical model, we standardize and normalize the pentad mean precipitation data $Y_{s,t}$ over Region $s$ at the lead time of $t$ to $V_{s,t}$ using the log-sinh transformation method proposed by Wang

et al. (2012). The zero values of pentad mean precipitation are treated as censored data. The selected preceding potential predictors $\boldsymbol{X}_{s,t}^{T} = [X_{1,s,t}\ X_{2,s,t} \cdots X_{n,s,t}]$ are standardized and normalized to $\boldsymbol{U}_{s,t}^{T} = [U_{1,s,t}\ U_{2,s,t} \cdots U_{n,s,t}]$ through the Yeo-Johnson transformation method as the input variables are allowed to be negative (Yeo and Johnson, 2000). The normalization parameters are estimated using the SCE-UA (shuffled complex evolution method developed at The University of Arizona) method that maximize the log-likelihood function. A more detailed





description of the log-sinh normalization method can be found in Wang et al. (2012).

There are many versions and variations of BHMs. In this study ,we establish the BHM model following Devineni et al. (2013) and Chen et al. (2014). However, the spatial correlation of precipitation over different regions is not considered here. A traditional no-pooling BHM is built for each hydroclimatic region separately. Meanwhile, the

potential predictors have been selected using the LASSO and stepwise regression previously. This indicates that the predictors used for establishing the BHM are independent. Thus, it is reasonable to model the related regression coefficients in the BHM independently. The transformed pentad mean precipitation $V_{s,t}$ is assumed to follow the normal distribution,

$$V_{s,t} \sim N(\mu_{s,t}, \sigma_{s,t}^2) \tag{4}$$

We then link the parameter $\mu_{s,t}$ with the normalized predictors using a linear model,

$$\mu_{s,t} = \beta_{0,s,t} + \sum_{i=1}^{n} \beta_{i,s,t} U_{i,s,t} \tag{5}$$

where $\beta_{i,s,t}$ is the slope term for Region $s$ at the lead time of $t$ corresponding to the predictor $U_{i,s,t}$. This indicates that the regression coefficients are modeled independently.

To complete the hierarchical formulation, we assume the unknown parameters, including $\sigma_{s,t}$, $\beta_{0,s,t}$, $\cdots$, $\beta_{n,s,t}$, follow non-informative priors:

$$\frac{1}{\sigma_{s,t}^2} \sim U(0, 100) \tag{6}$$

$$\beta_{i,s,t} \sim N(0, \ 10^4), \qquad i = 0, \cdots, n \tag{7}$$

This implies that the information used for posterior distribution inference is only provided by the data.


Given $\boldsymbol{\theta} = \{(\sigma_{s,t}, \ \beta_{i,s,t}), \ i = 0, \cdots, n\}$ denotes parameters in the Bayesian hierarchical model for Region $s$ at the lead time $t$, $V$ denotes the normalized pentad mean precipitation data, and $\boldsymbol{U}$ denotes the normalized preceding predictors, the full posterior of the parameters is given as:

$$p(\boldsymbol{\theta}|\boldsymbol{U}, V) \propto p(V|\boldsymbol{\theta}, \boldsymbol{U})p(\boldsymbol{\theta}) \tag{8}$$

where $p(V|\boldsymbol{\theta}, \boldsymbol{U})$ is the likelihood, and $p(\boldsymbol{\theta})$ is the prior of parameters $\boldsymbol{\theta}$. As the posterior distributions of parameters $\boldsymbol{\theta}$ are not standard distributions, it is difficult to conduct analytical integration. In this study, we use the R package *runjags* (Denwood, 2016) to calibrate the parameters of the BHM. The *runjags* offers an interface to facilitate calibrating BHMs employ a Gibbs sampling algorithm in Just Another Gibbs sampler (JAGS). The initial





values of model parameters $\boldsymbol{\theta}$ are first randomly sampled from prior distributions. The parameters $\boldsymbol{\theta}$ are then

updated based on the full conditional distributions. We use five independent Markov chains in each model run, with

a total number of 10, 000 iterations for each chain. The convergence is ensured by the potential scale reduction

factor $\hat{R}$ (Brooks and Gelman, 1998). An approximate convergence is diagnosed when the $\hat{R}$ is less than 1.1 for

all parameters.

Once the parameters are sampled, the Bayesian hierarchical model can be used to forecast sub-seasonal

precipitation using preceding large-scale circulation signals. Given new preceding predictors $\boldsymbol{X}_{s,t}^* = [X_{1,s,t}^* \cdots X_{n,s,t}^*]^T$

at the lead time of $t$, the normalized predictors $\boldsymbol{U}_{s,t}^* = [U_{1,s,t}^* \cdots U_{n,s,t}^*]^T$ are found using the estimated transformation

parameters during the training period. The posterior predictive distribution of normalized pentad mean precipitation

is given as:

$$V_{s,t}^* \sim N(\mu_{s,t}^*, \sigma_{s,t}^2) \tag{9}$$

$$\mu_{s,t}^* = \beta_{0,s,t} + \sum_{i=1}^n \beta_{i,s,t} U_{i,s,t}^* \tag{10}$$

Again, the Gibbs sampling algorithm is used to obtain samples of $V_{s,t}^*$ by given parameter value sets $\theta$. The

samples of $V_{s,t}^*$ are then back-transformed to produce ensemble precipitation forecasts of $Y_{s,t}^*$ over Region $s$ at

the lead time of $t$.

**2.2.4   Deterministic and Probabilistic Evaluation**

The deterministic sub-seasonal precipitation forecast skills are evaluated using the Kling-Gupta Efficiency (KGE):

$$\text{KGE} = 1 - \sqrt{(r-1)^2 + (\beta-1)^2 + (\gamma-1)^2} \tag{11}$$

$$r = \frac{\sum_{i=1}^N (y_i - \bar{y})(o_i - \bar{o})}{\sqrt{\sum_{i=1}^N (y_i - \bar{y})^2} \sqrt{\sum_{i=1}^N (o_i - \bar{o})^2}} \tag{12}$$

$$\beta = \frac{\mu_y}{\mu_o} \tag{13}$$

$$\gamma = \frac{\sigma_y}{\sigma_o} \tag{14}$$

where $y_i$ is the ensemble mean forecasts for case $i$, $i = 1,2,\cdots,N$; $o_i$ is the corresponding observation. $\mu_y$ is

the forecast mean for all cases; while $\mu_o$ is the observation mean for all cases. $\sigma_f$ is the standard deviation in

ensemble mean forecasts; while $\sigma_o$ is the standard deviation in observations. $r$ represents the correlation

coefficient between ensemble mean forecasts and the observations, $\beta$ represents the forecast bias, and $\gamma$

measures the variability error. Compared with other evaluation metrics, the KGE offers an insight into the model

performance as the decomposition into correlation, bias, and variability term. A full discussion of the KGE-statistics





sees Gupta et al. (2009) and Kling et al. (2012). The KGE ranges from negative infinity to one. A value of one suggests that the ensemble mean forecasts are the same as the observations.

The Continuous Ranked Probability Score (CRPS) is used to provide an overall evaluation of the accuracy of probabilistic sub-seasonal precipitation forecasts:

$$\text{CRPS} = \frac{1}{N}\sum_{i=1}^{N}\int[F_i(y) - H(y - o_i)]^2 dy \qquad (15)$$

where $F_i()$ is the cumulative distribution function of the ensemble forecasts for case $i$; and $H()$ is the Heaviside step function defined as:

$$H(y - o_i) = \begin{cases} 0 & y < o_i \\ 1 & y \ge o_i \end{cases} \qquad (16)$$

where $o_i$ is the corresponding observation.

The CRPS skill score is then calculated by comparing the CRPS of ensemble forecasts with the CRPS of reference forecasts:

$$\text{CRPS}_{SS} = \frac{\text{CRPS}_{REF} - \text{CRPS}}{\text{CRPS}_{REF}} \times 100\% \qquad (17)$$

The reference forecasts are generated using the Bayesian hierarchical model with no predictors used for prediction. A skill score of 100% indicates that the ensemble forecasts are the same as the observations, whereas a skill score of 0% suggests that the ensemble forecasts show no improvement over the cross-validated climatology. A negative skill score means that the ensemble forecasts are inferior to the cross-validated climatology.


The attribute diagram is used here to evaluate the reliability, resolution, and sharpness of the probabilistic forecasts. The attribute diagram shows the observed frequencies against its forecast probabilities for a given event with binary outcomes (Hsu and Murphy, 1986). In this study, the three class events of below-, near-, and above normal is defined by equally dividing the cross-validated climatology into terciles. The forecast probability is binned as 5

equal-width intervals. The corresponding observed relative frequency is plotted against the mean forecast probability in each bin. The forecasts are reliable if the scatters are along the 45-degree diagonal. The sharpness is also shown on the attribute diagram. The forecasts are sharp if the probabilities tend to be either very high (e.g. > 90%) or very low (e.g. <10%) (Peng et al., 2014). The sharpness is indicated by the size of dots in each bin. The

attribute diagram requires a large number of samples to draw robust conclusions. In this study, the probabilistic

forecasts over the 17 hydroclimatic regions are pooled together to increase the sample size for each lead time.

## 3.  Results

### 3.1 Deterministic forecast skill

Figure 3 summaries the KGE values for all regions and all lead times. Skillful deterministic sub-seasonal

precipitation forecasts are observed mainly in regions in southeastern and southwestern China. The KGE values

are above 0.2 over Region 2 (Inland Rivers in northern Tibet), Region 9 (Upper Yangtze River), Region 11

(Southwest Rivers in Southern Tibet), Region 12 (Southwest Rivers in Yunnan), Region 13 (Yangtze River), and

Region 16 (Pearl River) for almost all lead times. Although the KGE values are lower in Region 1 (Inland Rivers in

Xinjiang), Region 7 (Songhua River), Region 14 (Middle Yangtze River), Region 15 (Lower Yangtze River), and

Region 17 (Southeast Rivers), positive KGE values still can be found when the lead time is beyond 10 days. This

indicates that the deterministic forecasts still can provide useful in formation at longer lead times over these regions.

Much lower predictive skills are observed in northern and northeastern regions, that is Region 3 (Inland Rivers in

Inner Mongolia), Region 4 (Yellow River), Region 5 (Upper Yellow River), Region 6 (Hai River), Region 8 (Liao

River), and Region 10 (Huai River). The KGE values over these regions are near or below zero when the lead time

is beyond 5 days.

Figure 4 shows the ensemble mean forecasts during the period of 1979~2016, alongside 50% and 95% confidence

intervals from the Bayesian hierarchical model over Region 1 (Inland Rivers in Xinjiang) at different lead times. The

positive KGE values suggest that the established Bayesian hierarchical model can provide skillful deterministic

precipitation forecasts up to 25-days ahead for Region 1. The correlation coefficient $r$ is always above 0.35, the

biases are below 20% for all lead times. However, we should also note that the variability ratio $\gamma$ is below 0.5 when

the lead time is beyond 5 days. This suggests that the observed precipitation variability is underestimated for

ensemble mean forecasts.

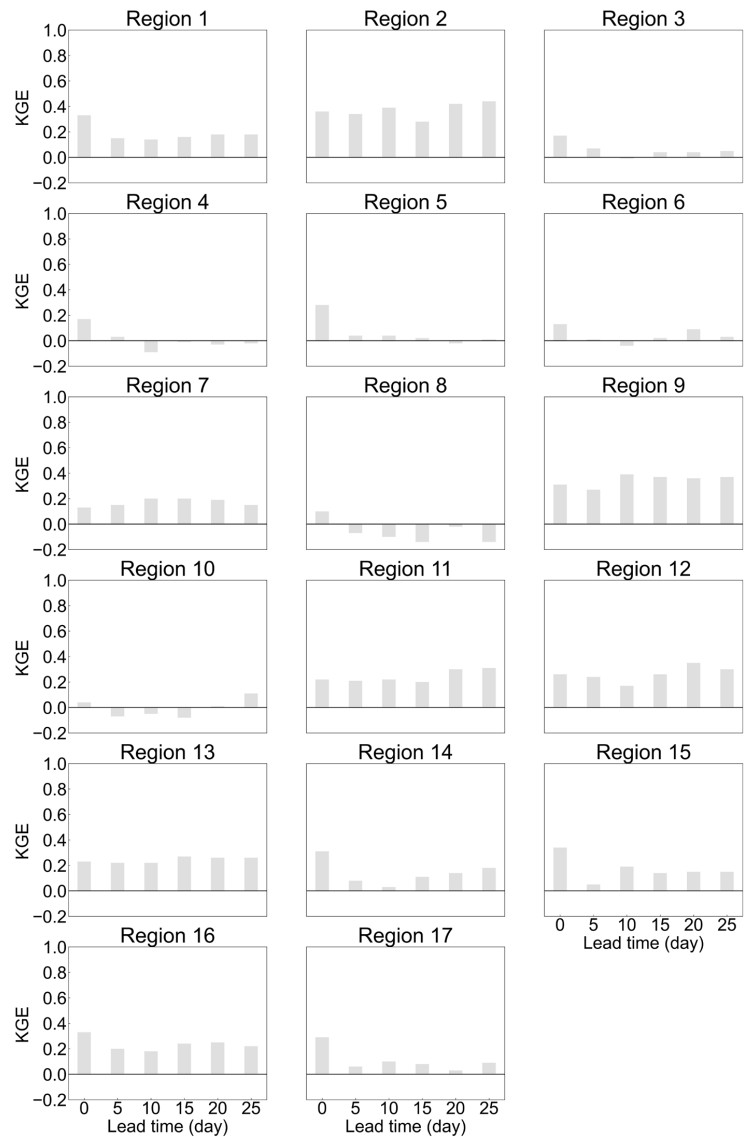


Figure 3. Kling-Gupta Efficiency of Pentad mean precipitation forecasts at different lead times over 17 hydroclimatic

regions during the boreal summer monsoon.



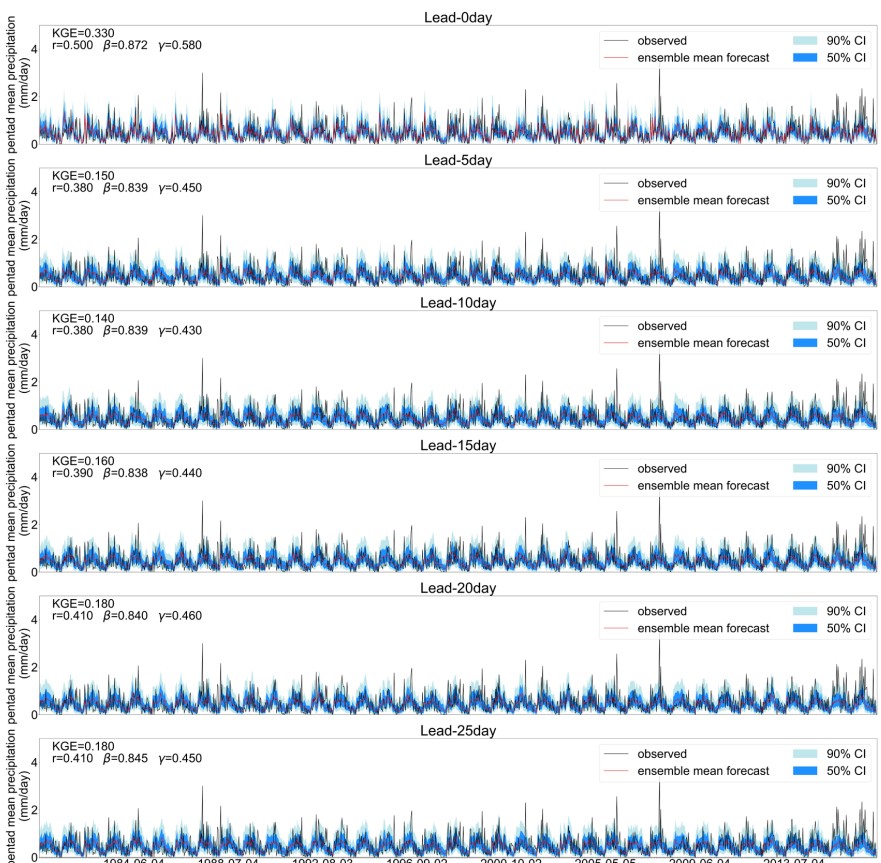

Figure 4. Pentad mean precipitation forecasts at different lead times over Region 1 (Inland Rivers in Xinjiang) during the boreal summer monsoon from 1979 to 2016. The ensemble mean forecasts are shown by the red line, observations by the black line, alongside 50% (shaded in blue) and 95% (shaded in powderblue) confidence intervals. CI = confidence interval.

**3.2 Probabilistic forecast skill**

The positive CRPS skill scores shown in Figure 5 suggest that the Bayesian hierarchical model is able to provide skillful probabilistic forecasts for all regions and all lead times. The CRPS skill scores are mostly over 10% over regions where positive KGE values are observed, including Region 2 (Inland Rivers in northern Tibet), Region 9 (Upper Yangtze River), Region 11 (Southwest Rivers in Southern Tibet), Region 12 (Southwest Rivers in Yunnan), Region 13 (Yangtze River), and Region 16 (Pearl River). Although the KGE values are negative over Region 3



(Inland Rivers in Inner Mongolia), Region 4 (Yellow River), Region 5 (Upper Yellow River), Region 6 (Hai River), Region 8 (Liao River), and Region 10 (Huai River) when the lead time is beyond 5 days, the positive CRPS skill scores suggest that the probabilistic forecasts still can provide valuable information compared to climatological forecasts at longer lead times.

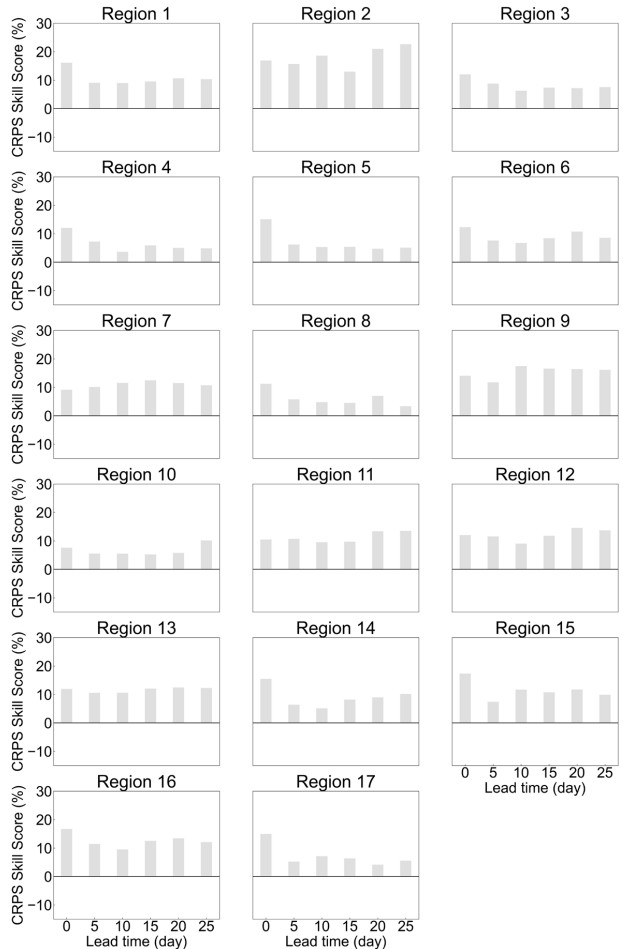

Figure 5. Continuous Ranked Probability Skill Score of Pentad mean precipitation forecasts at different lead times over 17 hydroclimatic regions during the boreal summer monsoon.

Attribute diagrams of the sub-seasonal precipitation probabilistic forecasts at different lead times are shown in Figure 6. Most points fall near the 1:1 line for all events and all lead times, indicating that the probabilistic forecast distributions are reliable. The results also suggest that the probabilistic forecasts are sharp at all lead times,



especially for below-normal and above normal categories.

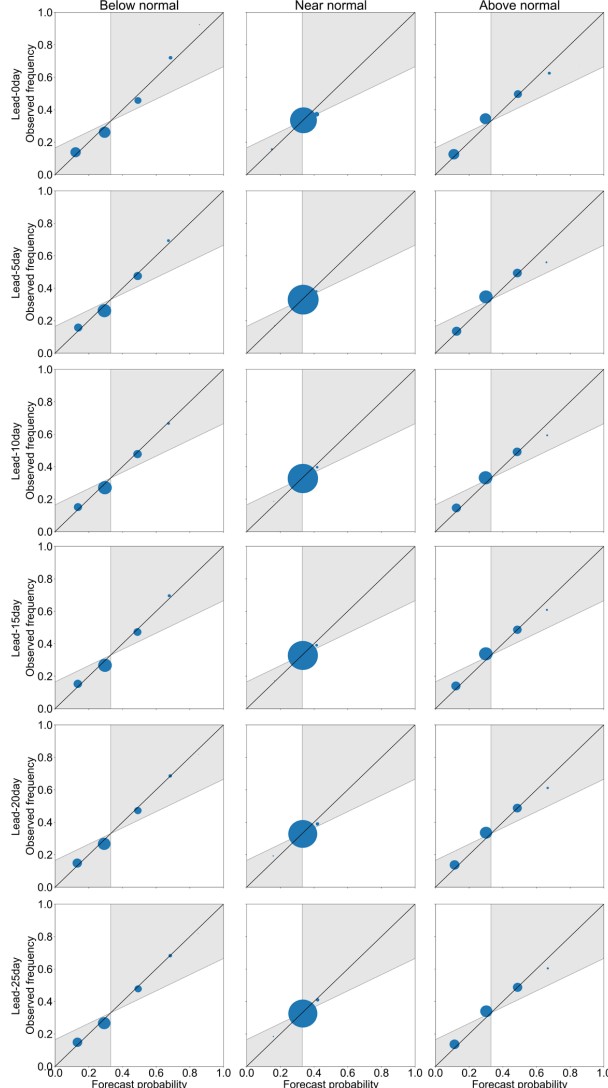

Figure 6. Attribute diagram of pentad mean precipitation forecasts during the boreal summer monsoon for tercile

based categories. Forecast probability is binned with width of 0.2, and the size of the dots indicates the sharpness

of probabilistic forecasts.



## 4. Discussion

### 4.1 Forecast skill and possible mechanism

In this study, we first define potential predictors by analyzing the relations between pentad mean precipitation and preceding 10-60-day ISO large-scale circulation signals. Robust predictors are identified using the Lasso regression and stepwise regression approaches. A Bayesian hierarchical model is then established to predict sub-seasonal precipitation for each hydroclimatic region. Our results demonstrate that the Bayesian hierarchical model could provide skillful deterministic sub-seasonal precipitation forecasts over southeastern and southwestern

hydroclimatic regions in China. However, the deterministic predictive skills over northeastern China are much lower. The decomposition of KGE values suggest that the intraseasonal variability is underestimated in these regions. This may be explained by the different characteristics of intraseasonal variability and different possible mechanism over different hydroclimatic regions. Wang (2007) analyzed the precipitation variability from April to September over China, and the results suggested that the seasonal component accounted for nearly 70% of the total variability

over northeastern China. The intraseasonal (10-90 days) component only accounted for nearly 7% of the total variability, which indicate that the intraseasonal precipitation over these regions have no significant frequency peak. In comparison, the sub-seasonal component accounted for over 20% of the total variability in southeastern and southwestern China. Ouyang and Liu (2020) also found that the boreal summer monsoon intraseasonal variability of precipitation over the lower Yangtze River basin was mainly dominated by the relatively low-frequency 12-20-

day variability and high-frequency 8-12-day variability. Wang and Duan (2015) demonstrated that the Quasi-biweekly oscillation (QBWO, 10-20 days oscillation) was the dominant mode of intraseasonal variability of summer precipitation over the Tibetan Plateau. The relations between atmospheric intraseasonal oscillation and the low-frequency variability of precipitation vary from region to region as well. Ren and Shen (2016) suggested that the impact of tropical atmospheric intraseasonal oscillation on precipitation were more significant in regions in southern

China and the Tibetan Plateau areas during the boreal summer.

Compared to deterministic forecasts, the probabilistic forecasts are more promising, especially at longer lead times. The CRPS skill scores are over 10% in southeastern and southwestern China when the lead time is beyond 10 days. The skill scores still reach a value of over 5% in other hydroclimatic regions. In contrast, the CRPS skill scores

of the BJP calibrated sub-seasonal precipitation forecasts were almost 0% at the lead time of 10-30 days (Li et al., 2020). This suggests that the Bayesian hierarchical model could provide useful forecast information at longer lead





times when preceding ISO signals are used as predictors. However, we should also note that the highest CRPS skill scores of the Bayesian hierarchical model are lower than 30% when the lead time is between 0-10 days, while that of the calibrated forecasts are over 50%. This indicates that the calibrated forecasts are more skillful for short

to medium range precipitation forecasts. The Calibration, Bridging, and Merging (CBaM) method, which makes the best use of GCM outputs, has been proved to be efficient for improving seasonal precipitation over many regions (Strazzo et al., 2019; Schepen and Wang, 2013; Peng et al., 2014). Recently, Specq and Batté (2020) proposed a similar statistical-dynamical approach to improve sub-seasonal precipitation forecasts over the southwest tropical Pacific. In the future, the statistical forecasts generated from lagged atmospheric indices should be included in the

calibrated forecasts to further improve sub-seasonal precipitation forecast skills.

**4.2 Limitations and future work**

In this study, the potential predictors are identified using the whole record in spite of the cross-validation strategy used for statistical modelling. This may introduce artificial skill into the models to some extent. However, defining

potential predictors for each step of the cross-validation is difficult in practice. The potential predictors were defined by averaging signals in the areas of significant correlations with large extent for each hydroclimatic region and lead time in this study. Nevertheless, this procedure was time consuming. Although we identified the irregular boundaries of significant correlated areas automatically by the Python package scikit-image (Walt et al., 2014), we should note the correlation did not equal causation. We carefully selected potential predictors by analyzing the possible

mechanism of sub-seasonal precipitation for each hydroclimatic region and each lead time. In addition, the definition of potential predictors for each step of the cross-validation is likely to yield similar results. Here, we analyzed the spatial patterns of correlations between lagged signals and filtered precipitation over Region 1 at the lead time of 0-day for each step of the leave-one-year out cross-validation. The results showed little variability compared to the correlation patterns shown in Figure 2. The cross-validation strategy used in the statistical

modelling procedure could also reduce the chance of overfitting (Vehtari and Lampinen, 2002; Delsole and Shukla, 2009).

Another limitation of this study is the treatment of zero values adopted in the statistical modelling procedure. We treated the zero values as censored data, which was also referred as "explicit" approach in Mcinerney et al. (2019).

Although this treatment performed well in "low-ephemeral" and "mid-ephemeral" catchments, the performance of

this "explicit" approach was poor in "high-ephemeral" (> 50% zero flows) catchments. Further development is required to overcome this problem. The copula functions are flexible in choosing marginal distributions, and have been widely used in hydrological simulations in recent years (Zhang and Singh, 2007; Vernieuwe et al., 2015; De Michele and Salvadori, 2003). Compared to the Bayesian statistics we used in this study, the copula functions are
more general and the normalization may not be required when the skewed distributions are used as the marginal distribution of precipitation. This may provide a possible solution to overcome the problems caused by the large amount of zero values.

We built the Bayesian hierarchical model for each hydroclimatic region separately. However, the spatial patterns of precipitation have not been considered yet. The spatial Bayesian hierarchical model, which is able to capture the
spatial dependence of precipitation between different regions, could be used to provide sub-seasonal precipitation forecasts with spatial coherence (Reza Najafi and Moradkhani, 2013; Bracken et al., 2016). An alternative way to reconstruct the spatial patterns of probabilistic precipitation forecasts is to use the Schaake Shuffle method or Ensemble Copula Coupling method (Roman et al., 2013; Clark et al., 2004). Higher spatial or temporal resolution
of precipitation forecasts are also needed for sub-seasonal streamflow forecasts. However, our previous studies indicated that post-processed daily precipitation forecasts from GCMs are of low accuracy when the lead time is beyond 10-14 days. In this study, the large-scale ISO signals was only used to predict pentad mean precipitation as the daily precipitation noise was too large.    Spatial or temporal disaggregation may be required in the future to provide daily precipitation forecasts as inputs for hydrological models.


## 5.  Conclusions

Sub-seasonal precipitation forecasts are difficult as the predictability from atmospheric initialization is lost after two weeks, while the slowly varying boundary conditions do not have substantial impact at such a time scale. The intraseasonal oscillation (ISO) is considered as one of the leading sources of sub-seasonal predictability. However,
the relationships between ISO signals and precipitation are of high uncertainty. In this study, we first defined potential predictors by analyzing the correlations between preceding ISO signals and precipitation for each hydroclimatic region and each lead time. The LASSO and stepwise regression approaches were used to narrow down the number of potential predictors. A Bayesian hierarchical model was then established to predict sub-seasonal precipitation during the boreal summer monsoon season. The model performance was evaluated through

a leave one-year out cross-validation strategy for both deterministic and probabilistic forecasts.

Our results suggested that the statistical model we built in this study could provide skillful deterministic sub-seasonal precipitation forecasts over southeastern and southwestern hydroclimatic regions at a lead time of 20-25 days. However, the deterministic forecast skills are much lower over northeastern China, partly owing to the

underestimation of intraseasonal variability in these regions. The probabilistic forecasts are more promising, especially at longer lead times. The skill scores and attribute diagrams demonstrated that the statistical model was able to provide skillful and reliable sub-seasonal precipitation forecasts for all hydroclimatic regions from 0-day to 25-day leads compared to climatological forecasts. This suggests that the probabilistic forecasts still could provide useful information by addressing the uncertainties of relationships between ISO signals and precipitation at sub-

seasonal time scales.

In this study, the large-scale circulation ISO signals were extracted from the zonal wind at 850 and 200 hPa, Outgoing Longwave Radiation, and the geopotential height at 850, 500, and 200 hPa. Other sources of sub-seasonal predictability, such as soil moisture, snow cover, and stratosphere-troposphere interaction, could be

included in the Bayesian hierarchical model to further improve sub-seasonal precipitation forecasts. The Calibration, Bridging, and Merging (CBaM) method could also be investigated at sub-seasonal time scale to further improve the forecast skills (Schepen and Wang, 2013; Schepen et al., 2014).

**Data availability**

The precipitation dataset used in this study can be derived from http://www.gloh2o.org/mswep/. The outgoing Longwave radiation (OLR) dataset can be found at https://www.ncdc.noaa.gov/news/new-outgoing-longwave-radiation-climate-data-record, and the ERA5 dataset can be sourced from https://cds.climate.copernicus.eu/.

**Author contribution**

Y.L. and Z.Y. WU designed the experiments and Y.L. carried them out. H.H preprared the data, and H.Y. developed the model code and performed the simulations. Y.L. prepared the manuscript with contributions from all co-authors.



**Competing interests**

The authors declare that they have no conflict of interest.


**Acknowledgements**

This work was funded by the National Natural Science Foundation of China (Grant number 52009027), the

Fundamental Research Funds for the Central Universities (Grant number 2019B10214).



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
