# Peer review of "Probabilistic sub-seasonal precipitation forecasts using preceding atmospheric intraseasonal signals in a Bayesian perspective"

_Hydrology and Earth System Sciences, 2022_

## Referee Comment (RC3)

**Review of manuscript HESS-2022-67 entitled "Subseasonal precipitation forecasts using preceding atmospheric intraseasonal oscillation signals in a Bayesian perspective" by Yuan Li, Zhiyong Yu, Hai He and Hao Yin**

**OVERALL RECOMMENDATION**

Major revision

**SUMMARY**

This manuscript proposes the construction of subseasonal forecasts of pentad-mean precipitation in 17 hydroclimatic regions in China. These subseasonal forecasts are generated using a purely statistical method called Bayesian Hierarchical Modeling (BHM). This statistical model uses previously observed climate variables at the global scale as input. The performance of the statistical subseasonal forecasts is evaluated with a deterministic metric (Kling-Gupta Efficiency) and probabilistic diagnostics (CRPSS, reliability diagram). The authors show that, in terms of probabilistic verification, these forecasts are still skillful in all 17 regions up to 25-day lead time. They also claim that the most interesting results are to be found in southern China.

**MAJOR COMMENTS**

This is a very relevant topic to propose statistical models for subseasonal forecasting based on lagged relationships. Not only can they be used as a benchmark to assess dynamical subseasonal forecasts (e.g S2S, SubX), but they might also prove more skillful than them. This seems to be the underlying claim of the authors for the statistical forecasts in the manuscript.
Then, I consider this study might be worthy of publication. However, it suffers from a lack of details that cast a doubt on the real added value of the method. I therefore ask the authors to convince me of its benefits through major revisions, as I feel the main claims are insufficiently supported in the current version.

Here are my major concerns, by order of importance:

1) The scores that are used to claim the benefits of the method **should be compared** to the scores obtained with raw dynamical subseasonal forecasts (e.g ECMWF), and possibly with your own BJP-processed from Li et al (2020). All those scores should appear simultaneously in Figures 3 and 4.

2) The methodology should be illustrated with more figures besides Figure 2. For instance, you could show the results of the LASSO predictor selection for the Figure 2 example (Region 1) at a specific lead time. Then, you could also select a specific target week (e.g your May 1-May, 5, 1979 period) and simultaneously visualize the values of the different predictors and the predicted precipitation.

More generally speaking, my recommendation is to **open the "black box"** and give more visual information showing what the statistical model is doing and why it works.

3) A figure **summarizing the different steps of the statistical prediction** is necessary for the reader to have a complete vision of the workflow.

4) Some **spatial visualization of the scores** is missing, e.g a map where the 17 regions are colored according to their score. This is important to support the claim that the method performs best in southern China. You could also give names to the regions and indicate them on Figures 3 and 5, this would help a lot.

5) In order to compensate for the necessary additional details required in my comments 1 to 4, some parts of the manuscript **could be shortened** (e.g Introduction, Sections 4 and 5).

6) Section 2.2.4, l.316: "The reference forecasts are generated using the Bayesian hierarchical model with no predictors used for prediction."
l.318: "show no improvement over the cross-validated climatology"
→ **It is unclear to me what the reference in CPRSS is**. Is it the cross-validated climatology or the forecasts generated with no predictor? Are they the same? If so, you should state it explicitly.

**MINOR COMMENTS**

Figures 3 and 4: I think the graphical aspect of these figures could be improved (e.g vertical scale, colored bars, etc.).

Figure 4: The curves on Figure 4 are illegible as there are too many time steps. Personally, I can't see the red curve (model) and how it compares to the observations in blue. Actually, I'm not sure this figure is really necessary beyond the indications in the top left-hand corner (KGE, r, etc.), I suggest replacing by a table.

Figure 5: I am surprised that CRPSS does not decrease monotonically with lead time. Admittedly there can be some noisy variations at longer lead times, but I still find that some results are quite weird (e.g in Region 2, CRPSS at 20 days is better than at 0 day). Isn't there an effect of the reference that is used in the CRPSS? Some explanations should be provided.

l.414-416: Please specify what are "the BJP calibrated sub-seasonal precipitation forecasts" from Li et al. (2020). I guess it corresponds to post-processed outputs of dynamical subseasonal forecasts with a GCM, but you should remind it and give the name of the model. More generally, your assertions concerning the comparison between BHM and your previous method from Li et al (2020) should be illustrated more extensively (see Major Comment #1).

l.436-438: *"Here,* we analyzed the spatial patterns of correlations between lagged signals and filtered precipitation over Region 1 at the lead time of 0-day for each step of the leave-one-year out cross-validation".
I can't see where the results you are referring to are, e.g I don't know what "Here" stands for in this sentence.

l. 19: "owing to the *underestimation* of intraseasonal variability in this region"
l.396: "The decomposition of KGE values suggest that the intraseasonal variability is *underestimated* in these regions"
I am not sure "underestimation" is the correct word in this context. From what I understand, the important fact is that intraseasonal variability is of limited importance in those regions because it does not account for a large fraction of total variability, so the model cannot perform well in those regions. I suggest rephrasing.

l.381-382: "The results also suggest that the probabilistic forecasts are sharp at all lead times, especially for below-normal and above normal categories".
Judging by the reliability diagrams, I am not convinced by the sharpness of the forecasts. On the contrary, I think the authors should mention very limited sharpness. I guess this is intrinsic to a Bayesian approach relying on a non-informative prior.

**LANGUAGE AND TYPOS**

l.9: "as predictor**s**" → "as predictor"
l. 19: "owing to the underestimation of intraseasonal variability in this region". Why underestimation?
l.22: "Other sources (…)  **will** be included"
l. 22: "forecast skill**s**" → "forecast skill".
I think that the word "skill" is never expected to be plural in this context. Same remark at l.34, l.74, l.116 (x2), l.395, l.425
l.25: "mitigation**s**" → "mitigation"
l.28: "" → "launched"
l.30: "" → "cannot"
l. 32: "before it  **can** be used"
l.41: "atmospheric-oceanic indices" → Do you mean "atmospheric or oceanic indices"?
l.43: "dominant" → I suggest using another word, what about "more performant"?
l.45: "plenty **of**"
l.48-51: "a new cluster-based empirical method (…), which the sea surface temperature (…) were included as predictors.".
The sentence is unclear, I suggest rephrasing, e.g splitting the sentence in two: "a new cluster-based method (…) European and Mediterranean regions**. This method uses** sea surface temperature (…) as predictors".
l.56: "at such a time scale" Unnecessary, please remove.
l.69: "but **in** extra-tropical regions as well"

l.77-78: "the relationships between ISO signals and precipitation are of high uncertainty for different regions at different lead times". I suggest rephrasing, e.g "the relationships between ISO signals and precipitation **are highly uncertain and depend on the region and lead time.**"

l.79-81: "To our best knowledge, the uncertainties of relationships between preceding ISO signals and sub-seasonal precipitation have not been fully considered in sub-seasonal precipitation forecasts in previous studies." I suggest another formulation.

l.84: Remove the CSC acronym. You never use it in the rest of the article.

l.87: "Bayes-theorem based statistical models" → "**Bayesian** statistical models"

l. 91: Idem

l.104: "is frequently influenced by" → "is frequently **subject to**"

l.111: "the model performance (…)  **is** evaluated"

l.115-116: "the deterministic and probabilistic forecast **skill is** presented"

l.127: "is area-weighted averaged over 17 hydroclimatic regions"

l.134: "to monitor" → "to monitor"

l.139: "proved to be capable of reflecting the MJO structure as the zonal wind" Unclear → "proved to be **as** capable of reflecting the MJO structure as the zonal wind"?

l.148: "calculating efficiency" → "computational efficiency"?

l. 194, l.196: " Africa" → "Africa"

l. 234: "in (Nardi and Rinaldo, 2011; Mcneish, 2015)". Typo, remove parentheses.

l.301:  → "For a full description of KGE-statistics, see Gupta et al (2009)..."

---

## Author Comment (AC1)

*General comment*

The authors established a Bayesian hierarchical model (BHM) to predict the 10-60d precipitation for 17 hydroclimatic regions over China during the boreal summer monsoon season (May to October) by using the previous atmospheric intraseasonal signals. Both deterministic and probabilistic evaluations showed that the BHM provides skillful subseasonal forecasts over southeastern and southwestern hydroclimatic regions at a lead time of 20-25 days while the skills are poor over northeastern China, owing to the underestimation of intraseasonal variability.

The authors have conducted numerous calculations and employed many different statistical analysis methods. However, the explanation for their choice of the calculation and methods are deficient. Moreover, I cannot tell whether the BHM proposed in this paper show any superior skills than other statistical models or even dynamical S2S models. From this point of view, I incline to reject the manuscript, but I give an opportunity to the authors to improve the manuscript.

The authors thank the referee's valuable comments. As introduced in the introduction section, several statistical models have been developed to generate sub-seasonal precipitation forecasts. The Spatial-Temporal Projection Model (STPM), which extracts the coupled patterns of preditors and predictand, has been widely used in recent years (Hsu et al., 2020; Zhu and Li, 2017a, b, c, 2018). The STPM1 is based on the singular value decomposition (SVD) analysis, while the STPM2 is constructed by analyzing the spatial-temporal coupled co-variance patterns between predictors and predictand (Fig. 1). A more detailed description of STPM1 and STPM2 can be found in Hsu et al. (2015).

[Figure]

Fig. 1. Major steps of STPM1 and STPM2 prediction model (Hsu et al., 2015).

However, we should note that the uncertainty of sub-seasonal precipitation forecasts may be underestimated in STPM models. In previous studies, an optimal ensemble (OE) strategy was applied to pick up best predictors and generate probabilistic forecasts (Zhu and Li, 2017a; Zhu et al., 2015). Nevertheless, the ensemble number (number of best predictors) was always limited. Further statistical assumptions were required to inteprete limited ensembles as probabilistic forecasts. Compared to OE-based probabilistic forecasts, the Bayes-theorem based statistical models are more flexible and more efficient for assessing multiple sources of uncertainties. The Bayes-theorem based models have been widely used for various aspects, and the predictive probability distributions could be generated through Markov chain Monte Carlo sampling algorithms. Thus, we will develop a **STPM2-BHM** probabilistic forecast model by taking full advantages of both STPM and Bayesian statistical modelling. We will no longer define potential predictors by averaging ISO signals in the areas of significant correlations. Instead, the predictors will be defined by extracting the coupled patterns between pentad precipitation anomalies and atmospheric intraseasonal oscillation signals, which is also known as **STPM2** in Fig. 1. The **BHM** model is then built to address the **parameter uncertainty** in the **transfer function** shown in Fig. 1.

the following changes in Sect. 2.2.2 and Sect. 2.2.3 will be made:

In section 2.2.2, we defined potential predictors by averaging U850, U200, OLR, H850, H500, H200 signals in the areas of significant correlations at different lead times. The number of potential predictors was then narrowed down using the Least Absolute Shrinkage and Selection Operator (LASSO) regression and stepwise regression approaches.

In the revised manuscript, the LASSO and stepwive regression approaches will not be used to define potential predictors. Instead, we will follow the steps of STPM2 (Fig. 1) to define coupled pattern projection coefficients as predictors. In addition, the predictand Y will be the pentad mean precipitation anomalies as suggested by the referee. We first construct spatial-temporal coupled covariance patterns (COV) where the predictand Y (10-60-day component of precipitation) and predictors X (10-60-day component of ISO signals) are significantly correlated,

$$cov(Y,X) = \frac{1}{T}\sum_{t=1}^{T}(y_t - E(y))(x_t - E(x)) \tag{1}$$

where $t$ is the number of pentads during the training period, $y_t$ is the ISO of precipitation, and $x_t$ is the ISO of atmospheric fields.

The coupled pattern projection coefficient $X_p$ , which can also be regarded as the predictor, is then obtained by multiplying the covariance and predictors X, and summing the product for each grid point where a 95 % significant level is reached,

$$X_p = cov(Y,X) \times X \tag{2}$$

Comapred to previous studies, we will build a bayesian hierarchical model to address the parameter uncertainty of the transfer function $Y_p = \alpha X_p + \beta$ shown in Fig. 1.

Here, we assume the normalized predictand $Y_p$ follows the normal distribution,

$$Y_p \sim N(\mu_p, \sigma_p^2) \tag{3}$$

We then link the parameter $\mu_p$ with the normalized coupled pattern projection coefficient $X_p$ using a linear model,

$$\mu_p = \alpha X_p + \beta \tag{4}$$

To complete the hierarchical formulation, we assume the unknown parameters, including $\sigma_p$, $\alpha$, and $\beta$ follow non-informative priors:

$$\frac{1}{\sigma_p^2} \sim U(0, 100) \tag{5}$$

$$\alpha \sim N(0, \ 10^4) \tag{6}$$

$$\beta \sim N(0, \ 10^4) \tag{7}$$

The posterior distributions of these parameters will be obtained using the Markov chain Monte Carlo algorithm as well.

Fig. 2 presents the leave-one-year-out cross-validated sub-seasonal forecasts in 1981 as an example of the **STPM2-BHM** model. Here, the spatial-temporal coupled covariance patterns are derived from 10-60-day component of precipitation and U850. The predictand is the pentad mean anomalies of precipitation over Region 1 (Inland Rivers in Xinjiang). The **STPM2-BHM** model shows high forecast skills at different lead times, and  prediction skills are mainly come from intraseasonal component of U850 filed.

[Figure]

Fig. 2. Sub-seasonal forecasts of pentad mean precipitation anomalies over Region 1 (Inland Rivers in Xinjiang) during the boreal summer monsoon in 1981. The ensemble mean of STPM2-BHM forecasts are shown by the red line, observations by the black line, alongside 50% (shaded in blue) and 95% (shaded in powderblue) confidence intervals. CI = confidence interval.

We would like to incorporate the above proposals in the revised manuscript. A more detailed analysis of the results will be given to ensure that the **STPM2-BHM** model is reliable and robust for generating probabilistic forecasts of pentad mean precipitation anomalies over China.

*Major comments:*

1. The intraseasonal variability and the intraseasonal oscillation are different terms. The authors focus on the prediction of intraseasonal precipitation (10-60d) over China during summer (May to October). Although the selected predictors are atmospheric intraseasonal signals, no specific BSISO or MJO pattern can be found in the previous correlation maps. The title may be more consistent with the content after removing "oscillation".

Thanks for this comment. We will incorporate this suggestion in the revised manuscript.

2. The selected intraseasonal signals and the physical processes of their influencing on precipitation over China should be provided.

Thanks for this comment. We will provide the intraseasonal signals and physical processes of their influencing on precipitation over China as supplementary file in the revised manuscript.

3. For each region and each pentad from May to October, a BHM is built to forecast precipitation at different lead time. The detail information should be shown in caption of Fig.2, Fig.3, Fig.5. Are the results in these figures for a specific pentad or the average mean from May to October? If the latter is the case, will the skill for each pentad be similar throughout the whole summer?

Thanks for this comment. The results shown in Fig. 2, Fig. 3, and Fig. 5 in the manusctipt are the overall forecast skills by pooling all forecasts and observations from 1979~2016 together. We also agree that the forecast skills should be different for each pentad as the impacts of physical processes on precipitation vary at different time. Fig. 3 gives an example of the correlation coefficients between the ensemble mean of **STPM2-BHM model** forecasts and obervations for each pentad during the boreal summer monsoon over Region 1. Overall, the correlations show great diversity at different pentads from May to October. A more comprehensive analysis would be given in the revised manuscript to address this comment.

[Figure]

Fig. 3. Correlation coefficients between ensemble mean of STPM2-BHM model forecasts and the observations over Region 1 (Inland Rivers in Xinjiang). The predictors are obtained by analyzing the spatial-temporal coupled covariance patterns between ISO of precipitation and U850.

4. Figure 1 shows the division of the hydroclimatic regions. However, this is not a scientific way to divide China with respect to rainfall variation. Does the precipitation in each region have the coherent intraseasonal variation? If not, the correlation map is meaningless because they are calculated based on the areal-mean precipitation. Moreover, do we really need 17 regions?

The authors appreciate this suggestion. We agree that the intraseasonal variation of rainfall vary in different parts of China. Zhu and Li (2017a) used the rotated empirical orthogonal function (REOF) metod to divide the entire China into 10 sub regions as shown in Fig. 4. However, we would like to keep the division of 17 hydroclimatic regions in the revised manuscript for several reasons. We admit that the division proposed by Zhu and Li (2017a) could ensure that the precipitation in each region have coherent intraseasonal variation. However, this division may be difficult for other applications,

especially for hydrological modelling purpose. In this study, the sub-seasonal precipitation forecasts for each hydroclimatic region could be potentially used as inputs of conceptual hydrologic models to generate sub-seasonal streamflow forecasts. Meanwhile, the division of 17 hydroclimatic regions is based on both watershed division standard and climate classifications. This will ensure that the climatic characteristics are nearly uniform in each region. A more detailed description of the division could be found in Lang et al. (2014). To resolve this comment in the revised manuscript, we will highlight the reason why we choose to divide China into 17 hydroclimatic regions in Sect. 2 Data and Methodology.

[Figure]

Fig. 4. The division of China based on REOF of the 10–80-day summer rainfall (a) regions in west of China (b) regions in east of China (Zhu and Li, 2017a).

[Figure]

Fig. 5. 17 hydroclimatic regions over China.

1) No evidences are provided to justify the advantage of this prediction model. Does this model have better performance than the ECMWF S2S model? Or the spatial-temporal projection models (STPM)? The authors need to make some comparison.

Thanks for this comment. As mentioned above, we will build a **STPM2-BHM model** by taking full advantages of both STPM and Bayesian statistical modelling. Compared to STPM2 model introduced in previous studies, the **STPM2-BHM model** could provide predictive density functions by addressing parameter uncertainties. Thus, we would like to focus on probabilistic forecast skills. We will provide a more detailed description of the **STPM2-BHM model** in Sect. 2. Meanwhile, our previous study used the Bayesian joint probability (BJP) approach to calibrate ECMWF S2S model forecasts at different spatiotemporal scales (Li et al., 2021). The results suggested that the probabilistic forecast skills were almost zero when the lead time was beyond 10 days. To resolve this comment, we will provide more discussion on the results of **STPM2-BHM model** and our previous work.

2) From Figure 4, I can see the prediction skills mainly came from the annual cycle (which is quite stable), rather than the pentad variation (the intraseasonal component). How about the skills if only anomaly precipitation is verified? I think the skill is very limited.

Thanks for this comment. We will revise the method section as mentioned above, and the pentad mean precipitation anomalies is defined as the predictand Y in the **STPM2-BHM model**. The results shown in Fig. 2 and Fig. 3 indicate that the **STPM2-BHM model** could provide skilful sub-seasonal precipitation forecasts. A more comprehensive evaluation will be given to ensure that the newly built **STPM2-BHM model** is skilful for forecasting sub-seasonal precipitation anomalies.

*Minor comments:*

1. Page 1 Line 9, "..." is ".".

Thanks for this comment. We will incorporate this suggestion in the revised manuscript.

2. Page 9 Line 208, the order of Fig. 2 is confusing for the reader to discern the evolution of intraseasonal atmospheric signals from Lead 25d to Lead 0d. The figure can be sliced to two figures with the first one showing the correlation between preceding U850, U200, OLR and 10-60d precipitation from Lead 25d to Lead 0d, and the second one showing the remaining H850, H500, and H200.

Thanks for this comment. We will incorporate this suggestion in the revised manuscript (Fig. 6 and Fig. 7).

[Figure]

Fig. 6. Correlation between preceding ISO signals of U850, U200, OLRA and 10-60-day component of precipitation over Region 1 (Inland Rivers in Xinjiang) at different lead times. Correlation coefficients statistically significant at the 5% level are shaded.

[Figure]

Fig. 7. Same as Fig. 6, but for H850, H500, and H200.

3. Page 15 Line 355, Fig. 3. The skill of Kling-Gupta Efficiency (KGE) in region 2, region 9 and region 12 increases with time, why? Could you please show r, β and γ before you show KGE? Because correlation coefficient and bias are the basic metric for forecast verification.

Thanks for this comment. The KGE values increase with lead time over Region 2, Region 9, and Region 12 is mostly owing to the potential predictors we defined in Sect. 2.2.2. Averaging U850, U200, OLR, H850, H500, H200 signals in the areas of significant correlations may lose some useful signals. In the revised manuscript, we will define the predictors by multiplying the co-variance field and each

predictor and sum the product for each grid point (at each lag) where a 95% significant level is reached as the STPM2. Meanwhile, the pentad mean precipitation anomalies will be treated as the predictand Y. The KGE may not be suitable for evaluating the forecast accuracy as the cross-validated observation mean of anomalies is nearly zero. We will provide correlation coefficient and bias of the ensemble mean of STP2-BHM model forecasts instead of the KGE in the revised manuscript.

4. Page 15 Line 355, Fig. 3. The prediction skill (KGE) of region 1 is the best in 17 regions, but in Fig. 4, the BHM model shows no skills for extreme events. Please explain the reason.

Thanks for this comment. The low forecast skills of the BHM model for extreme events is mainly owing to the definition of potential predictors. Instead, we will built a **STPM2-BHM model** in the revised manuscript. The predictors are defined by the coupled pattern projection coefficient. Fig. 2 suggests that the newly built STPM2-BHM model is capable of predicting extreme events. To resolve this comment, ranked probabilistic forecast skills at different percentiles will also be given to evaluate the forecast skills of **STPM2-BHM model** for extreme events.

5.Page 16 Line 365. What is the standard of efficient prediction in KGE and Continuous Ranked Probability Score (CRPS)? In the paper, the authors use "0.2" and "positive" as the standards, what is the reason?

Thanks for this comment. Positive KGE values are always used as indicative 'good' simulations in hydrological simulations (Knoben et al., 2019). A CRPS skill score of 100% indicates that the ensemble forecasts are the same as the observations, whereas a skill score of 0% suggests that the ensemble forecasts show no improvement over the cross-validated climatology. A negative skill score means that the ensemble forecasts are inferior to the cross-validated climatology. In the revised manuscript, we will use the correlation coefficient and bias to evaluate deterministic forecast skills of the **STPM2-BHM model**, the CRPS skill score will be used to provide an overall evaluation of probabilistic forecasts, and the RPS skill scores at different percentiles will be used to evaluate the probabilistic forecast skills of extreme events.

6.Page 18 Line 385. The prediction skill over northeast China is relatively lower than that over southeastern and southwestern China. Although the number of samples will be induced, the results of southeastern and southwestern China can better demonstrate the skill of BHM.

Thanks for this comment. We will incorporate this suggestion in the revised manuscript.

7.Page 18 Line 385. There is no caption of a detail description of the size of dots.

Thanks for this comment. The size of the dots indicates the fraction of forecasts in that probability bin. We will incorporate this suggestion in the revised manuscript.

8. Line 355, during the boreal summer monsoon season.

Thanks for this comment. We will incorporate this suggestion in the revised manuscript.

9. Line 55-70, So far, there are many statistical models for subseasonal prediction (some of them were already used in operational subseasonal prediction). The authors may want to read or cite the following publications, and make comparisons with their model.

Zhu Z., T. Li, P.-C. Hsu, J. He, 2015: A spatial-temporal projection model for extended-range forecast in the tropics. Clim. Dyn., 45(3), 1085-1098. doi: 10.1007/s00382-014-2353-8.

Zhu Z., T. Li, 2018: Extended-range forecasting of Chinese summer surface air temperature and heat waves. Clim. Dyn., 50(5-6), 2007-2021. doi: 10.1007/s00382-017-3733-7.

Zhu Z., T. Li, 2017: The statistical extended-range (10–30-day) forecast of summer rainfall anomalies over the entire China. Clim. Dyn., 48(1), 209-224. doi: 10.1007/s00382-016-3070-2.

Zhu Z., T. Li, 2017: Empirical prediction of the onset dates of South China Sea summer monsoon. Clim. Dyn., 48(5), 1633-1645. doi: 10.1007/s00382-016-3164-x.

Zhu Z., T. Li, 2017: Statistical extended-range forecast of winter surface air temperature and extremely cold days over China. Q. J. R. Meteor. Soc., 704(143), 1528-1538. doi: 10.1002/qj.3023.

Zhu Z., S. Chen, K. Yuan, Y. Chen, S. Gao, Z. Hua, 2017: Empirical subseasonal predicting summer rainfall anomalies over the middle and lower reaches of Yangtze River basin based on the atmospheric intraseasonal oscillation. Atmos., 8(10), 185. doi:10.3390/atmos8100185.

Zhu Z., T. Li, L. Bai, J. Gao, 2017: Extended-range forecast for the temporal distribution of clustering tropical cyclogenesis over the western North Pacific. Theor. Appl. Climatol., 130(3), 865-877. doi: 10.1007/s00704-016-1925-4.

Li W., P. Hsu, J. He, Z. Zhu, W. Zhang, 2016: Extended-range forecast of spring rainfall in southern China based on the Madden–Julian Oscillation. Meteorol. Atmos. Phys., 128(3), 331-345. doi: 10.1007/s00703-015-0418-9.

Thanks for this comment. We will revise the introduction section to have a more detailed description of recent progresses on subseasonal predictions.

10.     Line 75-80, "However, we should note that the relationships between ISO signals and precipitation are of high uncertainty for different regions at different lead times"

Yes, that is why in Zhu and Li (2017), they used REOF to divided the mainland China into 10 subregions based on the coherent nature of the 10-90 variation in each subregion. They predicted 10-30day predictand at once because considering the whole process of intraseasonal variability with the time-varying and spatial varying information. The authors may want to read the paper via the following link:

http://dqkxxb.cnjournals.org/dqkxxb/article/abstract/20200120
Thanks for this comment. We have read this article, and a **STPM2-BHM model** will be built to take advantages of both STPM and Bayesian modelling.

**Reference:**

Hsu, P.-c., Zang, Y., Zhu, Z., and Li, T.: Subseasonal-to-seasonal(S2S) prediction using the spatial-temporal projection model (STPM), Transactions of Atmospheric Sciences, 43, 212-224, 2020.

Hsu, P.-C., Li, T., You, L., Gao, J., and Ren, H.-L.: A spatial‐temporal projection model for 10–30 day rainfall forecast in South China, Climate Dynamics, 44, 1227-1244, 10.1007/s00382-014-2215-4, 2015.

Knoben, W. J. M., Freer, J. E., and Woods, R. A.: Technical note: Inherent benchmark or not? Comparing Nash–Sutcliffe and Kling–Gupta efficiency scores, Hydrol. Earth Syst. Sci., 23, 4323-4331, 10.5194/hess-23-4323-2019, 2019.

Lang, Y., Ye, A., Gong, W., Miao, C., Di, Z., Xu, J., Liu, Y., Luo, L., and Duan, Q.: Evaluating Skill of Seasonal Precipitation and Temperature Predictions of NCEP CFSv2 Forecasts over 17 Hydroclimatic Regions in China, Journal of Hydrometeorology, 15, 1546-1559, 10.1175/JHM-D-13-0208.1, 2014.

Li, Y., Wu, Z., He, H., Wang, Q. J., Xu, H., and Lu, G.: Post-processing sub-seasonal precipitation forecasts at various spatiotemporal scales across China during boreal summer monsoon, Journal of Hydrology, 598, 125742, https://doi.org/10.1016/j.jhydrol.2020.125742, 2021.

Zhu, Z. and Li, T.: The statistical extended-range (10–30-day) forecast of summer rainfall anomalies over the entire China, Climate Dynamics, 48, 209-224, 10.1007/s00382-016-3070-2, 2017a.

Zhu, Z. and Li, T.: Empirical prediction of the onset dates of South China Sea summer monsoon, Climate Dynamics, 48, 1633-1645, 10.1007/s00382-016-3164-x, 2017b.

Zhu, Z. and Li, T.: Statistical extended-range forecast of winter surface air temperature and extremely cold days over China, Quarterly Journal of the Royal Meteorological Society, 143, 1528-1538, https://doi.org/10.1002/qj.3023, 2017c.

Zhu, Z. and Li, T.: Extended-range forecasting of Chinese summer surface air temperature and heat waves, Climate Dynamics, 50, 2007-2021, 10.1007/s00382-017-3733-7, 2018.

Zhu, Z., Li, T., Hsu, P.-c., and He, J.: A spatial–temporal projection model for extended-range forecast in the tropics, Climate Dynamics, 45, 1085-1098, 10.1007/s00382-014-2353-8, 2015.

---

## Author Comment (AC2)

**Responses to Comments on "Sub-seasonal precipitation forecasts using preceding atmospheric intraseasonal oscillation signals in a Bayesian perspective" (Referee #2)**

Anonymous referee #2  reported on 11 May 2022.

**Our responses are in blue and proposed manuscript revisions underlined.**

*General comments:*

This is a very relevant topic to propose statistical models for subseasonal forecasting based on lagged relationships. Not only can they be used as a benchmark to assess dynamical subseasonal forecasts (e.g S2S, SubX), but they might also prove more skillful than them. This seems to be the underlying claim of the authors for the statistical forecasts in the manuscript.

Then, I consider this study might be worthy of publication. However, it suffers from a lack of details that cast a doubt on the real added value of the method. I therefore ask the authors to convince me of its benefits through major revisions, as I feel the main claims are insufficiently supported in the current version.

Thanks for your comprehensive review and recognition of the study contribution. The constructive comments will help us improve our manuscript after revision. We provide detailed responses to your comments and our proposed manuscript revisions in the subsequent sections.

*Major comments:*

The scores that are used to claim the benefits of the method **should be compared** to the scores obtained with raw dynamical subseasonal forecasts (e.g ECMWF), and possibly with your own BJP-processed from Li et al (2020). All those scores should appear simultaneously in Figures 3 and 4.

Thanks for this comment. We agree that it is important to compare the skill scores of the statistical model we built in this study and the raw dynamical models. However, we also note that the configurations of the statistical model are not the same as the raw dynamical models. Consider, for example, predicting pentad mean precipitation during the period of $1^{th}$ May and $5^{th}$ May 1979. In this case, the pentad mean ISO signals during the period of $26^{th}$ April and $30^{th}$ April 1979 are used to predict the pentad mean precipitation at a lead time of 0-day (could also be referred as a lag time of 0-day). The pentad mean precipitation and corresponding ISO signals during the period of 1980-2016 from May to October are pooled together to make parameter reference. On the other hand, the raw dynamical models are not always able to provide pentad mean precipitation forecasts for the same period of $1^{th}$-$5^{th}$ May 1979 as the hindcast initial time, hindcast period, and hindcast frequency are different (Table 1). The comparison may be unfair if the predictand of the statistical model and raw dynamical models are not the same.

To overcome this problem, we would like to compare our results with the NCEP model. Although the NCEP model is not the top scoring model for sub-seasonal precipitation forecasts (De Andrade et al., 2019), the hindcast frequency of the NCEP model makes it able to generate pentad mean precipitation forecasts for the same period as the statistical model from 1999 to 2010 (Table 1).

Table 1. Configuration of S2S model hindcasts

| S2S model | Time range (days) | Spatial resolution | Hindcast frequency | Hindcast period | Ensemble size | Ocean coupling |
|---|---|---|---|---|---|---|
| ECMWF* | 46 | Tco639/Tco319, L91 | 2/week | Past 20 years | 11 | Yes |
| **NCEP** | **44** | **T126, L64** | **Daily** | **1999-2010** | **4** | **Yes** |
| JMA | 33 | TL479/TL319, L100 | 3/month | 1981-2010 | 5 | No |
| KMA* | 60 | N216, L85 | 4/month | 1991-2010 | 3 | Yes |
| UKMO* | 60 | N216, L85 | 4/month | 1993-2016 | 7 | Yes |
| CNRM | 61 | T255, L91 | 2/month | 1993-2014 | 15 | Yes |
| ECCC* | 32 | 0.45°X0.45°, L40 | Weekly | 1998-2017 | 4 | No |
| ISAC | 31 | 0.75°X0.56°, L54 | Every 5 days | 1981-2010 | 5 | No |
| BOM | 62 | T47, L17 | 6/month | 1981-2013 | 33 | Yes |
| CMA | 60 | T106, L40 | Daily | 1994-2014 | 4 | Yes |
| HMCR* | 61 | 1.1°X1.4°, L28 | Weekly | 1985-2010 | 10 | No |

*Hindcasts are produced on the fly (model version is not fixed)

Moreover, we will develop a STPM2-BHM statistical model as suggested by the Anonymous Referee #1. We will no longer define potential predictors by averaging ISO signals in the areas of significant correlations. Instead, the predictors will be defined by extracting the coupled patterns between pentad precipitation anomalies and atmospheric intraseasonal oscillation signals, which is also known as STPM2 in Fig. 1 (Hsu et al., 2015). The BHM model is then built to address the parameter uncertainty in the transfer function (Fig. 1). A more detailed description of the newly built STPM2-BHM model can be found at https://hess.copernicus.org/preprints/hess-2022-67/hess-2022-67-AC1-supplement.pdf.

[Figure]

Fig. 1. Major steps of STPM1 and STPM2 prediction model (Hsu et al., 2015).

The predictors we defined above could be used to predict pentad mean anomalies and pentad mean precipitation amount as well. Our previous responses to comments of the Anonymous Referee #1 had shown that the STPM2-BHM model was capable of predicting pentad mean anomalies. Here, we use the same predictors, including the U850, U200, OLRA, H850, H500, and H200, to predict pentad mean precipitation over different hydroclimatic regions. The results will then be compared with the NCEP model.

Figure 2 compares the pentad mean precipitation forecasts of the NCEP model and the STPM2-BHM model over Region 1 (Inland rivers in Xinjiang). The predictors are firstly defined by the spatial-temporal coupled covariance patterns of U850, U200, OLRA, H850, H500, and H200, seperately. All covariance patterns are then pooled together to predict pentad mean precipitation. It is not surprise that the NCEP model outperforms

the STPM2-BHM model when the lead time is within 5-10 days. However, the STPM2-BHM model shows much higher forecast skill compared to the NCEP model at longer lead times. In addition, the ensemble spread of the NCEP model is too narrow to provide reliable sub-seasonal forecasts as the ensemble size of dynamical models is always limited. In comparison, the STPM2-BHM model is of much higher reliability compared to raw dynamical models.

[Figure]

Fig. 2. Comparison of pentad mean precipitation forecasts of the NCEP model and the STPM2-BHM model over Region 1 (Inland rivers in Xinjiang).

We will incorporate the above proposals in the revised manuscript. A more detailed comparison of the NCEP model and the STPM2-BHM model will be given to provide evidence of the added value of the statistical method we proposed in this study.

The methodology should be illustrated with more figures besides Figure 2. For instance, you could show the results of the LASSO predictor selection for the Figure 2 example (Region 1) at a specific lead time. Then, you could also select a specific target week (e.g your May 1-May, 5, 1979 period) and simultaneously visualize the values of the different predictors and the predicted precipitation.

More generally speaking, my recommendation is to **open the "black box"** and give more visual information showing what the statistical model is doing and why it works.

Thanks for this comment. As suggested by the Anonymous Referee #1, we will no longer define potential predictors by averaging ISO signals in the areas of significant correlations. The LASSO and stepwive regression approaches will not be used to select potential predictors. Instead, we will follow the steps of STPM2 (Fig. 1) to define coupled pattern projection coefficients as predictors. We first construct spatial-temporal coupled covariance patterns (COV) where the predictand Y (pentad mean anomalies or pentad mean precipitation amount) and predictors X (10-60-day component of atmospheric ISO signals) are significantly correlated,

$$cov(X_{i,p}, Y) = \frac{1}{T} \sum_{t=1}^{T} (y_t - E(y))(x_{i,p,t} - E(x_{i,p})) . \tag{1}$$

where $t$ is the number of pentads during the training period, $y_t$ is the pentad mean anomalies or pentad mean precipitation amount, and $x_{i,p}$ is $p^{th}$ ISO atmospheric field of U850, U200, OLRA, H850, H500, and H200 for significantly correlated grid $i$.

The coupled pattern projection coefficient $X_p$ , which can also be regarded as the predictor, is then obtained by multiplying the covariance and predictors X, and summing the product for each grid point where a 95 % significant level is reached,

$$X_p = \sum_{i=1}^{N} cov(X_{i,p}, Y) * X_{i,p} \tag{2}$$

where $N$ is the total number of grid points where a 95% significant level of correlation coefficient is reached. Thus, there will be only one predictor for each ISO atmospheric field of U850, U200, OLRA, H850, H500, and H200.

To open the "black box" of the STPM2-BHM model, we will compare the forecast skill of the STPM2-BHM model with different predictors. Figure 3 shows the KGE values of the NCEP model and the STPM2-BHM model with different predictors over Region 1 (Inland rivers in Xinjiang). It is clear that the ISO signals of U850, U200, and H850 fileds contributed most to the sub-seasonal precipitation forecasts over Region 1 (Inland rivers in Xinjiang). In contrast, the OLRA and H200 may have limited effects on predicting sub-seasonal precipitation forecasts over this region.

We will have a more comprehensive analysis of the results by comparing the forecast skill of the STPM2-BHM model with different predictors as mentioned above.

[Figure]

Fig. 3. Comparison of KGE values of the NCEP model and the STPM2-BHM model over Region 1 with different predictors (Inland rivers in Xinjiang).

A figure **summarizing the different steps of the statistical prediction** is necessary for the reader to have a complete vision of the workflow.

Thanks for this comment. We have summarized the major steps of the STPM2-BHM model in Figure 4. We will add this flow chart in the Data and methodology section.

[Figure]

Fig. 4. Flow chart representing the major components of the STPM2-BHM model.

Some **spatial visualization of the scores** is missing, e.g a map where the 17 regions are colored according to their score. This is important to support the claim that the method performs best in southern China. You could also give names to the regions and indicate them on Figures 3 and 5, this would help a lot.

We agree with the referee that the spatial map of forecast skill over 17 regions could help to realize the spatial pattern of sub-seasonal predictability. However, this would lead to large number of figures as we also compare our forecast skill using different predictors. To address this comment, we will plot the heatmap to have a better illustration of forecast skill as shown in Figure 5. The region names are also presented. The NCEP model shows higher forecast skill in most regions when the lead time is within 10 days. However, the

STPM2-BHM model performs better compared to the NCEP model at longer lead times in most regions, except Region 15 (Lower Yangtze River), Region 16 (Pearl River), and Region 17 (Southeast rivers).

[Figure]

Fig. 5. Comparison of KGE values of the NCEP model and the STPM2-BHM model over 17 hydrological regions at different lead times.

In order to compensate for the necessary additional details required in my comments 1 to 4, some parts of the manuscript **could be shortened** (e.g Introduction, Sections 4 and 5).

Thanks for this comment. We will incorporate this suggestion in the revised manuscript.

Section 2.2.4, l.316: "The reference forecasts are generated using the Bayesian hierarchical model with no predictors used for prediction." l.318: "show no improvement over the cross-validated climatology" **It is

**unclear to me what the reference in CPRSS is**. Is it the cross-validated climatology or the forecasts generated with no predictor? Are they the same? If so, you should state it explicitly.

Thanks for this comment. In this study, the reference forecasts are generated with no predictors. This is the same meaning as the cross-validated climatology, which the mean and standard deviation of the predictand is only determined by the cross-validated precipitation data. We will have a more detailed description of the reference forecasts in the Data and methodology section.

*MINOR COMMENTS*

Figures 3 and 4: I think the graphical aspect of these figures could be improved (e.g vertical scale, colored bars, etc.).

Thanks for this comment. We will plot the heatmap to have a better illustration of forecast skill as shown in Figure 5.

Figure 4: The curves on Figure 4 are illegible as there are too many time steps. Personally, I can't see the red curve (model) and how it compares to the observations in blue. Actually, I'm not sure this figure is really necessary beyond the indications in the top left-hand corner (KGE, r, etc.), I suggest replacing by a table.

Thanks for this comment. We will remove Figure 4 in the revised manuscript, and the curves will be provided as supplementary files. The results of KGE values will be presented as shown above in Figure 5.

Figure 5: I am surprised that CRPSS does not decrease monotonically with lead time. Admittedly there can be some noisy variations at longer lead times, but I still find that some results are quite weird (e.g in Region 2, CRPSS at 20 days is better than at 0 day). Isn't there an effect of the reference that is used in the CRPSS? Some explanations should be provided.

Thanks for this comment. It is true that the forecast skill decreases as lead time increases for dynamical models. This is also can be found in Figure 5 and our previous study (Li et al, 2021).

[Figure]

Fig. 6. Boxplot diagrams of CPRS skill scores of ECMWF raw ensemble forecasts (top) and the BJP calibrated forecasts (bottom) at different lead times during the boreal summer monsoon. (Li et al., 2021)

However, we should note that the STPM2-BHM model is a purely statistical model. The forecast skill of the STPM2-BHM model is mostly determined by the relationship between precipitation and atmospheric ISO signals. The concurrent relationship between precipitation and atmospheric/oceanic signals may not be as

strong as lagged signals. For example, Shukla et al. (2011) found that the Nino-3 index had strongest relationship with Indian Summer Monsoon Rainfall Index (ISMRI) with a lag of 5 sesasons (MAM). Thus, the forecast skill of ISMRI were found to be higher at a lag of 5 seasons compared to a lag of 4 seasons when using the Nino-3 index. This is also found by many other studies, which the relationship between precipitation and large scale circulation signals may be stronger at longer lags (Kirono et al., 2010; Piechota et al., 1998). Thus, it is not surpise that the skill scores of the STPM2-BHM model are higher at longer lead times (could also be referred as longer time lags). To address this comment, we will provide comprehensive explanations of the results and compare with previous statistical modelling studies in the discussion section.

[Figure]

Fig. 7. Correlation coefficient (r) between India Summer Monsoon Rainfall Index (ISMRI) and (a) Niño 1+ 2 index, (b) Niño 3 index, (c) Niño 3.4 index and (d) Niño 4 index with Niño indices lagging by 1–8 season(s) (Shukla et al., 2011).

l.414-416: Please specify what are "the BJP calibrated sub-seasonal precipitation forecasts" from Li et al. (2020). I guess it corresponds to post-processed outputs of dynamical subseasonal forecasts with a GCM, but you should remind it and give the name of the model. More generally, your assertions concerning the comparison between BHM and your previous method from Li et al (2020) should be illustrated more extensively (see Major Comment #1).

Thanks for this comment. Our previous study post-processed ECMWF model outputs using the Bayesian Joint Probability (BJP) approach. In this study, we will compare the STPM2-BHM model with the NCEP model as we introduced previously.

l.436-438: "Here, we analyzed the spatial patterns of correlations between lagged signals and filtered precipitation over Region 1 at the lead time of 0-day for each step of the leave-one-year out cross-validation". I can't see where the results you are referring to are, e.g I don't know what "Here" stands for in this sentence.

Thanks for this comment. We have analyzed the non cross validated correlation and cross validated correlation between ISO of U850 and precipitation over Region 1 in Figure 8. However, the results are not presented in the manuscript as the cross validated correlation could be generated for each year for the leave-one-year-out cross validation strategy. To address this comment, we will provide the cross validated correlation maps as supplementary files.

[Figure]

Fig. 8. Comparison of non cross validated correlation and cross validated correlation between preceding ISO signals of U850 and 10-60-day component of precipitation over Region 1 (Inland Rivers in Xinjiang) at a lag of 0-day.

l. 19: "owing to the underestimation of intraseasonal variability in this region"

l.396: "The decomposition of KGE values suggest that the intraseasonal variability is underestimated in these regions" I am not sure "underestimation" is the correct word in this context. From what I understand, the important fact is that intraseasonal variability is of limited importance in those regions because it does not account for a large fraction of total variability, so the model cannot perform well in those regions. I suggest rephrasing.

We agree with the referee that the intraseasonal variability is of limited importance in these regions. We will rephase these sentences in the revised manuscript.

l.381-382: "The results also suggest that the probabilistic forecasts are sharp at all lead times, especially for below-normal and above normal categories". Judging by the reliability diagrams, I am not convinced by the sharpness of the forecasts. On the contrary, I think the authors should mention very limited sharpness. I guess this is intrinsic to a Bayesian approach relying on a non-informative prior.

The authors agree with the reviewer that the Bayesian approaches may have difficulty in predicting extreme events when non-informative prior is used. The copula-based statistical approches will be used in the future to see whether the shapness of forecasts could be improved. To address this comment, we will provide more discussion on the sharpness of sub-seasonal forecasts in the Discussion section.

LANGUAGE AND TYPOS

l.9: "as predictors" → "as predictor"

Thank you for this comment. We will incorporate this suggestion in the revised manuscript.

l. 19: "owing to the underestimation of intraseasonal variability in this region". Why underestimation?

Thank you for this comment. We agree with the referee that the intraseasonal variability is of limited importance in these regions. We will rephase these sentences in the revised manuscript.

l.22: "Other sources (…) would will be included"

Thank you for this comment. We will incorporate this suggestion in the revised manuscript.

l. 22: "forecast skill" → "forecast skill".

I think that the word "skill" is never expected to be plural in this context. Same remark at l.34, l.74, l.116 (x2), l.395, l.425

Thank you for this comment. We will incorporate this suggestion in the revised manuscript.

l.25: "mitigations" → "mitigation"

Thank you for this comment. We will incorporate this suggestion in the revised manuscript.

l.28: "lunched" → "launched"

Thank you for this comment. We will incorporate this suggestion in the revised manuscript.

l.30: "could not" → "cannot"

Thank you for this comment. We will incorporate this suggestion in the revised manuscript.

l. 32: "before it could can be used"

Thank you for this comment. We will incorporate this suggestion in the revised manuscript.

l.41: "atmospheric-oceanic indices" → Do you mean "atmospheric or oceanic indices"?

Thank you for this comment. The atmospheric-oceanic indices mean atmospheric or oceanic indices here. We will incorporate this suggestion in the revised manuscript.

l.43: "dominant" → I suggest using another word, what about "more performant"? l.45: "plenty of"

Thank you for this comment. We will incorporate this suggestion in the revised manuscript.

l.48-51: "a new cluster-based empirical method (…), which the sea surface temperature (…) were included as predictors.".
The sentence is unclear, I suggest rephrasing, e.g splitting the sentence in two: "a new cluster-based method (…) European and Mediterranean regions. This method uses sea surface temperature (…) as predictors".

Thank you for this comment. We will incorporate this suggestion in the revised manuscript.

l.56: "at such a time scale" Unnecessary, please remove. l.69: "but in extra-tropical regions as well"

Thank you for this comment. We will incorporate this suggestion in the revised manuscript.

l.77-78: "the relationships between ISO signals and precipitation are of high uncertainty for different regions at different lead times". I suggest rephrasing, e.g "the relationships between ISO signals and precipitation are highly uncertain and depend on the region and lead time."

Thank you for this comment. We will incorporate this suggestion in the revised manuscript.

l.79-81: "To our best knowledge, the uncertainties of relationships between preceding ISO signals and sub-seasonal precipitation have not been fully considered in sub- seasonal precipitation forecasts in previous studies." I suggest another formulation.

Thank you for this comment. We will rephrase this sentence in the revised manuscript as follows: To our best knowledge, the uncertainty in relationship between preceding ISO signals and sub-seasonal precipitation has not been fully considered yet.

l.84: Remove the CSC acronym. You never use it in the rest of the article.

Thank you for this comment. We will incorporate this suggestion in the revised manuscript.

l.87: "Bayes-theorem based statistical models" → "Bayesian statistical models"

Thank you for this comment. We will incorporate this suggestion in the revised manuscript.

l. 91: Idem

Thank you for this comment. We will incorporate this suggestion in the revised manuscript.

l.104: "is frequently influenced by" → "is frequently subject to" l.111: "the model performance (…) are is evaluated"

Thank you for this comment. We will incorporate this suggestion in the revised manuscript.

l.115-116: "the deterministic and probabilistic forecast skill is presented" l.127: "is area-weighted averaged over 17 hydroclimatic regions"

Thank you for this comment. We will incorporate this suggestion in the revised manuscript.

l.134: "to monitoring" → "to monitor"

Thank you for this comment. We will incorporate this suggestion in the revised manuscript.

l.139: "proved to be capable of reflecting the MJO structure as the zonal wind" Unclear → "proved to be as capable of reflecting the MJO structure as the zonal wind"?

Thank you for this comment. We will incorporate this suggestion in the revised manuscript.

l.148: "calculating efficiency" → "computational efficiency"?

Thank you for this comment. We will incorporate this suggestion in the revised manuscript.

l. 194, l.196: "the Africa" → "Africa"

Thank you for this comment. We will incorporate this suggestion in the revised manuscript.

l. 234: "in (Nardi and Rinaldo, 2011; Mcneish, 2015)". Typo, remove parentheses. l.301: "A full discussion of the KGE-statistics sees Gupta et al (2009)…" → "For a full description of KGE-statistics, see Gupta et al (2009)..."

Thank you for this comment. We will incorporate this suggestion in the revised manuscript.

**References**

de Andrade, F. M., Coelho, C. A. S., and Cavalcanti, I. F. A.: Global precipitation hindcast quality assessment of the Subseasonal to Seasonal (S2S) prediction project models, Climate Dynamics, 52, 5451-5475, https://doi.org/10.1007/s00382-018-4457-z, 2019.

Hsu, P.-C., Li, T., You, L., Gao, J., and Ren, H.-L.: A spatial–temporal projection model for 10–30 day rainfall forecast in South China, Climate Dynamics, 44, 1227-1244, 10.1007/s00382-014-2215-4, 2015.

Kirono, D. G. C., Chiew, F. H. S., and Kent, D. M.: Identification of best predictors for forecasting seasonal rainfall and runoff in Australia, Hydrological Processes, 24, 1237-1247, https://doi.org/10.1002/hyp.7585, 2010.

Li, Y., Wu, Z., He, H., Wang, Q. J., Xu, H., and Lu, G.: Post-processing sub-seasonal precipitation forecasts at various spatiotemporal scales across China during boreal summer monsoon, Journal of Hydrology, 125742, 2021.

Piechota, T. C., Chiew, F. H. S., Dracup, J. A., and McMahon, T. A.: Seasonal streamflow forecasting in eastern Australia and the El Niño–Southern Oscillation, Water Resources Research, 34, 3035-3044, https://doi.org/10.1029/98WR02406, 1998.

Shukla, R. P., Tripathi, K. C., Pandey, A. C., and Das, I. M. L.: Prediction of Indian summer monsoon rainfall using Niño indices: A neural network approach, Atmospheric Research, 102, 99-109, https://doi.org/10.1016/j.atmosres.2011.06.013, 2011.

---

## Author Response (AR1)

**Editor decision:**

Dear authors,

Thank you for responding to the two reviews. You have responded to most comments carefully. Because the comments are substantial and both reviewers suggest major revisions before the manuscript can be reconsidered, your revised manuscript will be sent to the referees again.

Please revise the manuscript accordingly. I look forward to receiving your revised manuscript.

Sincerely,

Yi He, HESS Editor

We are grateful to you for the kind decision. We have conducted a thorough revision to improve the manuscript as suggested by the insightful and constructive comments of the reviewers. The point-by-point responses are provided in the following.

Anonymous Referee #1 Received and published on 28 March 2022.

Our responses are in blue and revisions are in blue and italics, with the reviewer's comments shown as normal text.

**General comment**

The authors established a Bayesian hierarchical model (BHM) to predict the 10-60d precipitation for 17 hydroclimatic regions over China during the boreal summer monsoon season (May to October) by using the previous atmospheric intraseasonal signals. Both deterministic and probabilistic evaluations showed that the BHM provides skillful subseasonal forecasts over southeastern and southwestern hydroclimatic regions at a lead time of 20-25 days while the skills are poor over northeastern China, owing to the underestimation of intraseasonal variability.

The authors have conducted numerous calculations and employed many different statistical analysis methods. However, the explanation for their choice of the calculation and methods are deficient. Moreover, I cannot tell whether the BHM proposed in this paper show any superior skills than other statistical models or even dynamical S2S models. From this point of view, I incline to reject the manuscript, but I give an opportunity to the authors to improve the manuscript.

The authors thank the referee's valuable comments. As introduced in the introduction section, several statistical models have been developed to generate sub-seasonal precipitation forecasts. The Spatial-Temporal Projection Model (STPM), which extracts the coupled patterns of predictors and predictand, has been widely used in recent years (Hsu et al., 2020; Zhu and Li, 2017a, b, c, 2018). The STPM1 is based on the singular value decomposition (SVD) analysis, while the STPM2 is constructed by analyzing the spatial-temporal coupled co-variance patterns between predictors and predictand (Figure S1). A more detailed description of STPM1 and STPM2 can be found in Hsu et al. (2015).

[Figure]

Figure S1. Major steps of STPM1 and STPM2 prediction model (Hsu et al., 2015).

However, we should note that the uncertainty of sub-seasonal precipitation forecasts may be underestimated in STPM models. In previous studies, an optimal ensemble (OE) strategy was applied to pick up best predictors and generate probabilistic forecasts (Zhu and Li, 2017a; Zhu et al., 2015). Nevertheless, the number of best predictors was always limited. Further statistical assumptions were required to interpret limited ensembles as probabilistic forecasts.

Compared to OE-based probabilistic forecasts, the Bayes-theorem based statistical models are more flexible and more efficient for assessing multiple sources of uncertainties. The Bayes-theorem based models have been widely used for various aspects, and the predictive probability distributions could be generated through Markov chain Monte Carlo sampling algorithms. Thus, we develop a STP-BHM probabilistic forecast model by taking full advantages of both STPM and Bayesian statistical modelling. We no longer define potential predictors by averaging ISO signals in the areas of significant correlations. Instead, the predictors are defined by extracting the coupled patterns between atmospheric intraseasonal oscillation signals and precipitation, which is also known as STPM2 in Fig. S1. The BHM model is then built to address the parameter uncertainty in the transfer function shown in Fig. S1.

We added the spatial-temporal projection part in the predictor definition section from **L. 251** to **L. 260** as follows:
*The spatial-temporal coupled co-variance patterns are then constructed for grid point where the correlation statistically significant at the 5% level. The predictor is then defined by summing the product of the co-variance patterns and ISO signals of atmospheric field at each preceding pentad,*

[revised manuscript text omitted]

We also agree that it is of great importance to compare the skill scores of STP-BHM model we built in this study and the raw dynamical models. However, we note that the configurations of the statistical model are not the same as the dynamical models. The dynamical models are not always able to provide pentad mean precipitation forecasts for the same period as the STP-BHM model because the hindcast initial time, hindcast period, and hindcast frequency are different (Table 1). The comparison may be unfair if the predictand of the statistical model and dynamical models are not the same.

To overcome this problem, we have compared our results of the STP-BHM model with the NCEP model in the S2S Database. Although the NCEP model is not the top scoring model for sub-seasonal precipitation forecasts (De Andrade et al., 2019), the hindcast frequency of the NCEP model makes it able to generate pentad mean precipitation forecasts for the same period as the BHM model from 1999 to 2010 (Table S1).

Table S1. Configuration of S2S model hindcasts

| S2S model | Time range (days) | Spatial resolution | Hindcast frequency | Hindcast period | Ensemble size | Ocean coupling |
|---|---|---|---|---|---|---|
| ECMWF* | 46 | Tco639/Tco319, L91 | 2/week | Past 20 years | 11 | Yes |
| **NCEP** | **44** | **T126, L64** | **Daily** | **1999-2010** | **4** | **Yes** |
| JMA | 33 | TL479/TL319, L100 | 3/month | 1981-2010 | 5 | No |
| KMA* | 60 | N216, L85 | 4/month | 1991-2010 | 3 | Yes |
| UKMO* | 60 | N216, L85 | 4/month | 1993-2016 | 7 | Yes |
| CNRM | 61 | T255, L91 | 2/month | 1993-2014 | 15 | Yes |
| ECCC* | 32 | 0.45°X0.45°, L40 | Weekly | 1998-2017 | 4 | No |
| ISAC | 31 | 0.75°X0.56°, L54 | Every 5 days | 1981-2010 | 5 | No |
| BOM | 62 | T47, L17 | 6/month | 1981-2013 | 33 | Yes |
| CMA | 60 | T106, L40 | Daily | 1994-2014 | 4 | Yes |
| HMCR* | 61 | 1.1°X1.4°, L28 | Weekly | 1985-2010 | 10 | No |

*Hindcasts are produced on the fly (model version is not fixed)

We added the reason for the choice of the NCEP model from **L. 165** to **L. 175** as follows:

*The STP-BHM model we built in this study is compared to the dynamical models to provide a benchmark for sub-seasonal precipitation forecasts. However, the dynamical models are not always able to provide pentad mean precipitation forecasts for the same period as the STP-BHM model as the hindcast initial time, hindcast period, and hindcast frequency are different. The comparison may be unfair if the predictand of the statistical model and dynamical models are not the same. To overcome this problem, we compare our results of the STP-BHM model with the NCEP model archived in the S2S Database for the same period of 1999-2010 from May to October (http://apps.ecmwf.int/datasets/data/s2s/). The NCEP hindcasts are produced by the Climate Forecast System version 2 (CFSv2), which is composed of land, ocean and atmosphere components. The system provides a 4-member ensemble run every day from 1st January 1999 to 31 December 2010. More*

*details of the NCEP hindcasts are available at https://confluence.ecmwf.int/display/S2S/NCEP+Model+Description. The pentad mean precipitation amount forecasts of the NCEP model are generated to be consistent with the STP-BHM model.*

We added the comparison of the STP-BHM model and the NCEP model from **L. 418** to **L. 426** as follows:
*Figure 9 compares the CRPS skill scores of the STP-BHM model and the NCEP model from May to October during the period of 1999~2010. Although the NCEP model is not the top scoring model for sub-seasonal precipitation forecasts, the hindcast frequency of the NCEP model makes it able to generate pentad mean precipitation forecasts for the same period as the STP-BHM model from 1999 to 2010. It is not surprise that the NCEP model outperforms the STP-BHM model when the lead time is within 5 days. However, we should note that the STP-BHM model shows much higher probabilistic forecast skill compared to the NCEP model at longer lead times. Positive CRPS skill scores are observed for the STP-BHM model over most hydroclimatic regions, whereas the skill scores are mostly negative for the NCEP model.*

[Figure]

*Figure 9. The comparison of the CRPS skill scores of the STP-BHM model and the NCEP model from May to October during the period of 1999~2010.*

**Major comments:**

1. The intraseasonal variability and the intraseasonal oscillation are different terms. The authors focus on the prediction of intraseasonal precipitation (10-60d) over China during summer (May to October). Although the selected predictors are atmospheric intraseasonal signals, no specific BSISO or MJO pattern can be found in the previous correlation maps. The title may be more consistent with the content after removing "oscillation".

Thanks for this comment. We have removed the word "oscillation", and the title has been justified as follows:

*Probabilistic sub-seasonal precipitation forecasts using preceding atmospheric intraseasonal signals in a Bayesian perspective*

2. The selected intraseasonal signals and the physical processes of their influencing on precipitation over China should be provided.

Thanks for this comment. We have provided the intraseasonal signals and physical processes of their influencing on precipitation over China as supplementary file in the revised manuscript in Figures S1 to S32.

3. For each region and each pentad from May to October, a BHM is built to forecast precipitation at different lead time. The detail information should be shown in caption of Fig.2, Fig.3, Fig.5. Are the results in these figures for a specific pentad or the average mean from May to October? If the latter is the case, will the skill for each pentad be similar throughout the whole summer?

Thanks for this comment. We have revised the figure captions to give more details of the results as follows:

**L. 235 to L. 237:** *Figure 3. Correlation coefficient between preceding pentad mean 10-60-day signals of U850, U200, OLRA and precipitation over Region 1 (Inland Rivers in Xinjiang) at different lead times during the period of 1979~2016 from May to October. Correlation coefficients statistically significant at the 5% level are shaded.*

**L. 378 to L. 379:** *Figure 5. The cross-validated CRPS skill scores of the STP-BHM model for pentad mean precipitation amount forecasts at different lead times during the period of 1979-2016 from May to October.*

**L. 390 to L. 392:** *Figure 6. The Brier skill scores of the STP-BHM model for the prediction of below-normal and above-normal events of pentad mean precipitation amount at different lead times during the period of 1979-2016 from May to October.*

We also agree that the forecast skill is different throughout the whole summer. Figure S2 gives an example of the correlation coefficients between the ensemble mean of STP-BHM model forecasts and observations for pentad mean precipitation anomalies over Region 1. Overall, the correlations show great diversity at different pentads from May to October. However, these results are beyond the main scope of this study. We will analyze the possible reasons of these diversities in the future.

[Figure]

Figure S2. Correlation coefficients between ensemble mean of STP-BHM model forecasts and the observations for pentad mean precipitation anomalies over Region 1 (Inland Rivers in Xinjiang). The predictors are defined by the ISO signals of atmospheric fields of U850, U200, OLRA, H850, H500, and H200.

4. Figure 1 shows the division of the hydroclimatic regions. However, this is not a scientific way to divide China with respect to rainfall variation. Does the precipitation in each region have the coherent intraseasonal variation? If not, the correlation map is meaningless because they are calculated based on the areal-mean precipitation. Moreover, do we really need 17 regions?

The authors appreciate this suggestion. We agree that the intraseasonal variation of rainfall vary in different parts of China. Zhu and Li (2017a) used the rotated empirical orthogonal function (REOF) method to divide the entire China into 10 sub regions as shown in Fig. S3. However, we would like to keep the division of 17 hydroclimatic regions in the revised manuscript for several reasons. We admit that the division proposed by Zhu and Li (2017a) could ensure that the precipitation in each region have coherent intraseasonal variation. However, this division may be difficult for other applications, especially for hydrological modelling purpose. In

this study, the sub-seasonal precipitation forecasts for each hydroclimatic region could be potentially used as inputs of conceptual hydrologic models to generate sub-seasonal streamflow forecasts. Meanwhile, the division of 17 hydroclimatic regions is based on both watershed division standard and climate classifications. This will ensure that the climatic characteristics are nearly uniform in each region. A more detailed description of the division could be found in Lang et al. (2014).

[Figure]

Figure S3. The division of China based on REOF of the 10–80-day summer rainfall (a) regions in west of China (b) regions in east of China (Zhu and Li, 2017a).

[Figure]

Fig. S4. 17 hydroclimatic regions over China.

1) No evidences are provided to justify the advantage of this prediction model. Does this model have better performance than the ECMWF S2S model? Or the spatial-temporal projection models (STPM)? The authors need to make some comparison.

Thanks for this comment. As mentioned above, we build a STP-BHM model by taking full advantages of both STPM and Bayesian statistical modelling. We have compared our results of the STP-BHM model with the NCEP model in the S2S Database. Although the NCEP model is not the top scoring model for sub-seasonal precipitation forecasts, the hindcast frequency of the NCEP model makes it able to generate pentad mean precipitation forecasts for the same period as the STP-BHM model from 1999 to 2010. The STP-BHM model shows much higher forecast skill compared to the NCEP model when the lead time is beyond 5 days (Figure

9 in the revised manuscript). Positive CRPS skill scores are observed over most hydroclimatic regions for the STP-BHM model, whereas the skill scores are mostly negative for the raw NCEP model.

[Figure]

*Figure 9. The comparison of the CRPS skill scores of the STP-BHM model and the NCEP model from May to October during the period of 1999~2010.*

2) From Figure 4, I can see the prediction skills mainly came from the annual cycle (which is quite stable), rather than the pentad variation (the intraseasonal component). How about the skills if only anomaly precipitation is verified? I think the skill is very limited.

Thanks for this comment. We have revised the methodology section, and a STP-BHM model is built to predict both pentad mean precipitation amount and pentad mean precipitation anomalies. The CRPS skill scores of the STP-BHM model are presented in Figure 5 and Figure 10. In addition, we also assess the capability of the STP-BHM model for predicting the above-normal and below-normal events. The Brier skill scores are presented in Figure 6 and Figure 11. Positive CRPS skill scores and Brier skill scores are found over almost all regions and all lead times, indicating that the STP-BHM model outperforms the cross-validated climatological forecasts.

[Figure]

Figure 5. The cross-validated CRPS skill scores of the STP-BHM model for pentad mean precipitation amount forecasts at different lead times during the period of 1979-2016 from May to October.

[Figure]

Figure 10. Same as Figure 5, but for pentad mean precipitation anomalies.

[Figure]

Figure 6. The Brier skill scores of the STP-BHM model for the prediction of below-normal and above-normal events of pentad mean precipitation amount at different lead times during the period of 1979-2016 from May to October.

[Figure]

Figure 11. Same as Figure 6, but for pentad mean precipitation anomalies.

**Minor comments:**

1. Page 1 Line 9, ".." is ".".

Thanks for this comment. The redundant period has been removed.

2. Page 9 Line 208, the order of Fig. 2 is confusing for the reader to discern the evolution of intraseasonal atmospheric signals from Lead 25d to Lead 0d. The figure can be sliced to two figures with the first one showing the correlation between preceding U850, U200, OLR and 10-60d precipitation from Lead 25d to Lead 0d, and the second one showing the remaining H850, H500, and H200.

Thanks for this comment. We have sliced Fig. 2 into two figures in the revised manuscript as follows:

[Figure]

Figure 3. Correlation coefficient between preceding pentad mean 10-60-day signals of U850, U200, OLRA and precipitation over Region 1 (Inland Rivers in Xinjiang) at different lead times during the period of 1979~2016 from May to October. Correlation coefficients statistically significant at the 5% level are shaded.

[Figure]

*Figure 4. Same as Fig. 3, but for H850, H500, and H200.*

3. Page 15 Line 355, Fig. 3. The skill of Kling-Gupta Efficiency (KGE) in region 2, region 9 and region 12 increases with time, why? Could you please show r, β and γ before you show KGE? Because correlation coefficient and bias are the basic metric for forecast verification.

Thanks for this comment. The Kling-Gupta Efficiency, correlation coefficient, and bias are mainly used to evaluate deterministic forecast skill. In the revised manuscript, we mainly focus on the probabilistic forecast skill of the STP-BHM model we built in this study. Thus, the CRPS skill score is used to evaluate the overall probabilistic forecast skill, while the Brier skill score is used to evaluate the model performance for predicting above-normal and below-normal events. The reliability of probabilistic forecasts is evaluated through the attribute diagram. The KGE, correlation coefficient, and bias metrics are no longer used in the revised manuscript.

4. Page 15 Line 355, Fig. 3. The prediction skill (KGE) of region 1 is the best in 17 regions, but in Fig. 4, the BHM model shows no skills for extreme events. Please explain the reason.

Thanks for this comment. We have revised the methodology section, and a STP-BHM model is built to predict pentad mean precipitation amount and pentad mean precipitation anomalies. The model performance of predicting extreme events is assessed through the Brier skill score for above-normal and below-normal events in Figure 6 and Figure 11. The results suggest that the newly built model can provide skillful forecasts for extreme events as well, and positive Brier skill scores are observed over all hydroclimatic regions and lead times.

[Figure]

Figure 6. The Brier skill scores of the STP-BHM model for the prediction of below-normal and above-normal events of pentad mean precipitation amount at different lead times during the period of 1979-2016 from May to October.

[Figure]

Figure 11. Same as Figure 6, but for pentad mean precipitation anomalies.

5. Page 16 Line 365. What is the standard of efficient prediction in KGE and Continuous Ranked Probability Score (CRPS)? In the paper, the authors use "0.2" and "positive" as the standards, what is the reason?

Thanks for this comment. Positive KGE values are always used as indicative 'good' simulations in hydrological simulations (Knoben et al., 2019). However, we mainly focus on probabilistic forecast skill in the revised manuscript. Thus, the KGE is no longer used in this study.

A CRPS skill score of 100% indicates that the ensemble forecasts are the same as the observations, whereas a skill score of 0% suggests that the ensemble forecasts show no improvement over the cross-validated climatology. A negative skill score means that the ensemble forecasts are inferior to the cross-validated climatology. Like the CRPS skill score, the Brier skill score takes the value 100% for perfect forecasts and 0% for the reference forecasts. Positive skill scores indicate that the forecast skill is higher than the cross-validated climatology.

6.Page 18 Line 385. The prediction skill over northeast China is relatively lower than that over southeastern and southwestern China. Although the number of samples will be induced, the results of southeastern and southwestern China can better demonstrate the skill of BHM.

Thanks for this comment. We have revised the manuscript, and the spatial patterns of skill scores also indicate that the STP-BHM model performs better in southern China.

7.Page 18 Line 385. There is no caption of a detail description of the size of dots.

Thanks for this comment. We have revised the caption of the attribute diagrams from **L. 413 to L. 416** as follows:

*Figure 8. The attribute diagram of the STP-BHM model for the prediction of below-normal and above-normal events of pentad mean precipitation amount at different lead times. Forecast probability is binned with width of 0.2. The size of each dot represents the fraction of forecasts that fall into a particular probability bin..*

8.      Line 355, during the boreal summer monsoon season.

Thanks for this comment. We have incorporated this suggestion in the revised manuscript.

9.      Line 55-70, So far, there are many statistical models for subseasonal prediction (some of them were already used in operational subseasonal prediction). The authors may want to read or cite the following publications, and make comparisons with their model.

Zhu Z., T. Li, P.-C. Hsu, J. He, 2015: A spatial-temporal projection model for extended-range forecast in the tropics. Clim. Dyn., 45(3), 1085-1098. doi: 10.1007/s00382-014-2353-8.

Zhu Z., T. Li, 2018: Extended-range forecasting of Chinese summer surface air temperature and heat waves. Clim. Dyn., 50(5-6), 2007-2021. doi: 10.1007/s00382-017-3733-7.

Zhu Z., T. Li, 2017: The statistical extended-range (10–30-day) forecast of summer rainfall anomalies over the entire China. Clim. Dyn., 48(1), 209-224. doi: 10.1007/s00382-016-3070-2.

Zhu Z., T. Li, 2017: Empirical prediction of the onset dates of South China Sea summer monsoon. Clim. Dyn., 48(5), 1633-1645. doi: 10.1007/s00382-016-3164-x.

Zhu Z., T. Li, 2017: Statistical extended-range forecast of winter surface air temperature and extremely cold days over China. Q. J. R. Meteor. Soc., 704(143), 1528-1538. doi: 10.1002/qj.3023.

Zhu Z., S. Chen, K. Yuan, Y. Chen, S. Gao, Z. Hua, 2017: Empirical subseasonal predicting summer rainfall anomalies over the middle and lower reaches of Yangtze River basin based on the atmospheric intraseasonal oscillation. Atmos., 8(10), 185. doi:10.3390/atmos8100185.

Zhu Z., T. Li, L. Bai, J. Gao, 2017: Extended-range forecast for the temporal distribution of clustering tropical cyclogenesis over the western North Pacific. Theor. Appl. Climatol., 130(3), 865-877. doi: 10.1007/s00704-016-1925-4.

Li W., P. Hsu, J. He, Z. Zhu, W. Zhang, 2016: Extended-range forecast of spring rainfall in southern China based on the Madden–Julian Oscillation. Meteorol. Atmos. Phys., 128(3), 331-345. doi: 10.1007/s00703-015-0418-9.

Thanks for this comment. We have read and cited the recent publications on sub-seasonal forecasts in the revised manuscript from **L. 79** to **L. 92** as follows:

*The spatial-temporal projection (STP) model, which extracts the coupled patterns of predictors and predictand, has been developed in recent years (Hsu et al., 2020; Zhu and Li, 2017a, b, c, 2018). Hsu et al. (2015) established a set of spatial-temporal projection models (STPMs) to predict sub-seasonal precipitation at a lead time of 10-30 days over southern China. Their results suggested that the forecast skill was still promising at a 20-25-day lead time. Zhu and Li (2017a) predicted sub-seasonal precipitation by constructing STPMs over entire China, and independent forecasts of rainfall anomalies during the period of Olympic Games in 2008 and Shanghai World Expo in 2010 suggested that the STPMs were able to reproduce intraseasonal rainfall patterns at a 20-day lead time. However, we should note that the relationship between ISO signals and precipitation is highly uncertain and depend on the region and lead time. In previous studies, an optimal ensemble (OE) strategy was applied to generate probabilistic forecasts by picking up best predictors (Zhu and Li, 2017a; Zhu et al., 2015). Nevertheless, the number of best predictors was always limited. Further statistical assumptions were required to interpret limited ensembles as probabilistic forecasts. The uncertainty in relationship between preceding ISO signals of atmospheric field and precipitation has not been fully considered yet.*

Meanwhile, we develop the STP-BHM model by taking full advantages of both the STP model developed by Zhu and Li (2017a, b, c, 2018) and Bayesian statistical modelling. The results suggest that the STP-BHM model can provide skillful and reliable probabilistic forecasts at sub-seasonal time scale.

10.     Line 75-80, "However, we should note that the relationships between ISO signals and precipitation are of high uncertainty for different regions at different lead times"

Yes, that is why in Zhu and Li (2017), they used REOF to divided the mainland China into 10 subregions based on the coherent nature of the 10-90 variation in each subregion. They predicted 10-30day predictand at once because considering the whole process of intraseasonal variability with the time-varying and spatial varying information. The authors may want to read the paper via the following link:

http://dqkxxb.cnjournals.org/dqkxxb/article/abstract/20200120

Thanks for this comment. We have developed the STP-BHM model by taking full advantages of both the STP model developed by Zhu and Li (2017a, b, c, 2018) and Bayesian statistical modelling.

Response to Comments on "Sub-seasonal precipitation forecasts using preceding atmospheric intraseasonal oscillation signals in a Bayesian perspective" (Referee #2)

Anonymous Referee #2 reported on 11 May 2022.

Our responses are in blue and revisions are in blue and italics, with the reviewer's comments shown as normal text.

**General comments:**

This is a very relevant topic to propose statistical models for subseasonal forecasting based on lagged relationships. Not only can they be used as a benchmark to assess dynamical subseasonal forecasts (e.g S2S, SubX), but they might also prove more skillful than them. This seems to be the underlying claim of the authors for the statistical forecasts in the manuscript.

Then, I consider this study might be worthy of publication. However, it suffers from a lack of details that cast a doubt on the real added value of the method. I therefore ask the authors to convince me of its benefits through major revisions, as I feel the main claims are insufficiently supported in the current version.

Thanks for your comprehensive review and recognition of the study contribution. The constructive comments will help us improve our manuscript after revision. We provide detailed responses to your comments and our revised manuscript in the subsequent sections.

**Major comments:**

The scores that are used to claim the benefits of the method **should be compared** to the scores obtained with raw dynamical subseasonal forecasts (e.g ECMWF), and possibly with your own BJP-processed from Li et al (2020). All those scores should appear simultaneously in Figures 3 and 4.

Thanks for this comment. We agree that it is of great importance to compare the skill scores of sub-seasonal forecasts of the statistical model we built in this study and the raw dynamical models. However, we also note that the configurations of the statistical model are not the same as the dynamical models. Consider, for example, predicting pentad mean precipitation during the period of 1st May and 5th May 1979. In this case, the pentad mean ISO signals during the period of 26th April and 30th April 1979 are used to predict the pentad mean precipitation at a lead time of 0-day. The pentad mean precipitation and corresponding ISO signals during the period of 1980-2016 from May to October are pooled together to make parameter reference for the same lead time. On the contrary, the S2S dynamical models are not always able to provide pentad mean precipitation forecasts for the same period of 1st-5th May 1979 as the hindcast initial time, hindcast period, and hindcast frequency are different (Table 1). The comparison may be unfair if the predictand of the statistical model and dynamical models are not the same.

To overcome this problem, we would like to compare our results with the NCEP model. Although the NCEP model is not the top scoring model for sub-seasonal precipitation forecasts (De Andrade et al., 2019), the hindcast frequency of the NCEP model makes it able to generate pentad mean precipitation forecasts for the same period as the BHM model from 1999 to 2010 (Table S1).

Table S1. Configuration of S2S model hindcasts

| S2S model | Time range (days) | Spatial resolution | Hindcast frequency | Hindcast period | Ensemble size | Ocean coupling |
|---|---|---|---|---|---|---|
| ECMWF* | 46 | Tco639/Tco319, L91 | 2/week | Past 20 years | 11 | Yes |
| **NCEP** | **44** | **T126, L64** | **Daily** | **1999-2010** | **4** | **Yes** |
| JMA | 33 | TL479/TL319, L100 | 3/month | 1981-2010 | 5 | No |
| KMA* | 60 | N216, L85 | 4/month | 1991-2010 | 3 | Yes |
| UKMO* | 60 | N216, L85 | 4/month | 1993-2016 | 7 | Yes |
| CNRM | 61 | T255, L91 | 2/month | 1993-2014 | 15 | Yes |
| ECCC* | 32 | 0.45°X0.45°, L40 | Weekly | 1998-2017 | 4 | No |
| ISAC | 31 | 0.75°X0.56°, L54 | Every 5 days | 1981-2010 | 5 | No |
| BOM | 62 | T47, L17 | 6/month | 1981-2013 | 33 | Yes |
| CMA | 60 | T106, L40 | Daily | 1994-2014 | 4 | Yes |
| HMCR* | 61 | 1.1°X1.4°, L28 | Weekly | 1985-2010 | 10 | No |

*Hindcasts are produced on the fly (model version is not fixed)

In addition, we develop a STPM2-BHM statistical model as suggested by the Anonymous Referee #1. We will no longer define potential predictors by averaging ISO signals in the areas of significant correlations. Instead, the predictors are defined by extracting the coupled patterns between pentad precipitation anomalies and atmospheric intraseasonal oscillation signals, which is also known as STPM2 in Fig. 1 (Hsu et al., 2015). The BHM model is then built to address the parameter uncertainty in the transfer function shown in Fig. S1.

[Figure]

Fig. S1. Major steps of STPM1 and STPM2 prediction model (Hsu et al., 2015).

We have added the spatial-temporal projection part in the predictor definition section from **L. 251** to **L. 260** as follows:

[revised manuscript text omitted]

We added the comparison of the STP-BHM model and the NCEP model from **L. 418** to **L. 426** as follows:
Figure 9 compares the CRPS skill scores of the STP-BHM model and the NCEP model from May to October during the period of 1999~2010. Although the NCEP model is not the top scoring model for sub-seasonal precipitation forecasts, the hindcast frequency of the NCEP model makes it able to generate pentad mean precipitation forecasts for the same period as the STP-BHM model from 1999 to 2010. It is not surprise that the NCEP model outperforms the STP-BHM model when the lead time is within 5 days. However, we should note that the STP-BHM model shows much higher probabilistic forecast skill compared to the NCEP model at longer lead times. Positive CRPS skill scores are observed for the STP-BHM model over most hydroclimatic regions, whereas the skill scores are mostly negative for the NCEP model.

[Figure]

*Figure 9. The comparison of the CRPS skill scores of the STP-BHM model and the NCEP model during the period of 1999~2010 from May to October.*

The methodology should be illustrated with more figures besides Figure 2. For instance, you could show the results of the LASSO predictor selection for the Figure 2 example (Region 1) at a specific lead time. Then, you could also select a specific target week (e.g your May 1-May, 5, 1979 period) and simultaneously visualize the values of the different predictors and the predicted precipitation. More generally speaking, my recommendation is to **open the "black box"** and give more visual information showing what the statistical model is doing and why it works.

Thanks for this comment. As suggested by the Anonymous Referee #1, we will no longer define potential predictors by averaging ISO signals in the areas of significant correlations. The LASSO and stepwise regression approaches will not be used to select potential predictors. A STP-BHM model is built to predict both pentad mean precipitation amount and pentad mean precipitation anomalies.

To open the "black box" of the STP-BHM model, we also establish the STP-BHM model for U850, U200, OLRA, H850, H500, and H200, separately. The forecast skill of the STP-BHM model with different predictors are compared in Figure 7 from **L. 394** to **L. 400**, and Figure 12 from **L. 451** to **L. 454** as follows:

*Figure 7 compares the CRPS skill scores of pentad mean precipitation forecasts with different predictors. In general, U850, U200, H850, and H500 show higher forecast skill compared to OLRA and H200 for almost all hydroclimatic regions and lead times. This suggests that the ISO signals of these atmospheric fields contribute more to the overall forecast skill. Compared to the STP-BHM model built with only one predictor, the forecast skill is further improved when all ISO signals of atmospheric fields are used.*

[Figure]

*Figure 7. The cross-validated CRPS skill scores for sub-seasonal forecasts of pentad mean precipitation amount with different predictors (U850, U200, OLRA, H850, H500, H200). ALL denotes that the ISO signals of all atmospheric fields are used as predictors.*

*Figure 12 compares the CRPS skill scores of pentad mean precipitation anomalies with different predictors. Overall, the STP-BHM model with OLRA used as predictor shows higher forecast skill compared to other predictors for almost all hydroclimatic regions and lead times. This suggests that the OLRA contributes most to the overall forecast skill of pentad mean precipitation anomalies.*

[Figure]

*Figure 12. Same as Figure 7, but for pentad mean precipitation anomalies.*

A figure **summarizing the different steps of the statistical prediction** is necessary for the reader to have a complete vision of the workflow.

Thanks for this comment. We have summarized the major steps of the STP-BHM model in Figure 2 in the revised manuscript as follows:

[Figure]

*Figure 2. workflow of the spatial-temporal projection based Bayesian hierarchical model (STP-BHM).*

Some **spatial visualization of the scores** is missing, e.g a map where the 17 regions are colored according to their score. This is important to support the claim that the method performs best in southern China. You could also give names to the regions and indicate them on Figures 3 and 5, this would help a lot.

Thanks for this comment. We have presented the spatial maps of skill scores in the revised manuscript, and region names are also added in the heatmaps as follows:

[Figure]

*Figure 5. The cross-validated CRPS skill scores of the STP-BHM model for pentad mean precipitation amount forecasts at different lead times during the period of 1979-2016 from May to October.*

[Figure]

*Figure 7. The cross-validated CRPS skill scores of the STP-BHM model for pentad mean precipitation amount forecasts with different predictors (U850, U200, OLRA, H850, H500, H200). ALL denotes that the ISO signals of all atmospheric fields are used as predictors.*

In order to compensate for the necessary additional details required in my comments 1 to 4, some parts of the manuscript **could be shortened** (e.g Introduction, Sections 4 and 5).

Thanks for this comment. We have shortened the corresponding sections mentioned above in the revised manuscript.

Section 2.2.4, I.316: "The reference forecasts are generated using the Bayesian hierarchical model with no predictors used for prediction." I.318: "show no improvement over the cross-validated climatology" **It is unclear to me what the reference in CPRSS is**. Is it the cross-validated climatology or the forecasts generated with no predictor? Are they the same? If so, you should state it explicitly.

Thanks for this comment. In this study, the reference forecasts are generated with no predictors. This is the same meaning as the cross-validated climatology, which the mean and standard deviation of predictand is only determined by the cross-validated precipitation data.

**MINOR COMMENTS**

Figures 3 and 4: I think the graphical aspect of these figures could be improved (e.g vertical scale, colored bars, etc.).

Thanks for this comment. We have replaced these figures by the spatial maps of skill scores in the revised manuscript as shown above.

Figure 4: The curves on Figure 4 are illegible as there are too many time steps. Personally, I can't see the red curve (model) and how it compares to the observations in blue. Actually, I'm not sure this figure is really necessary beyond the indications in the top left-hand corner (KGE, r, etc.), I suggest replacing by a table.

Thanks for this comment. In the revised manuscript, we mainly focus on the probabilistic forecast skill of the STP-BHM model. Thus, the KGE is no longer used for verification. The probabilistic skill scores and attribute diagrams are shown in the revised manuscript instead of KGE.

Figure 5: I am surprised that CRPSS does not decrease monotonically with lead time. Admittedly there can be some noisy variations at longer lead times, but I still find that some results are quite weird (e.g in Region 2, CRPSS at 20 days is better than at 0 day). Isn't there an effect of the reference that is used in the CRPSS? Some explanations should be provided.

Thanks for this comment. It is true that the forecast skill decreases as lead time increases for dynamical models. This is also can be observed in Figure 9 (the NCEP model) and our previous study (Li et al, 2020).

[Figure]

*Figure 9. The comparison of the CRPS skill scores of the STP-BHM model and the NCEP model from May to October during the period of 1999~2010.*

[Figure]

Fig. S4. Boxplot diagrams of CPRS skill scores of ECMWF raw ensemble forecasts (top) and the BJP calibrated forecasts (bottom) at different lead times during the boreal summer monsoon. (Li et al., 2021)

However, we should note that the STP-BHM model is a purely statistical model. The forecast skill of the STP-BHM model is mostly determined by the relationship between precipitation and atmospheric ISO signals. The concurrent relationship between precipitation and atmospheric/oceanic signals may not be as strong as lagged signals. For example, Shukla et al. (2011) found that the Nino-3 index had strongest relationship with Indian Summer Monsoon Rainfall Index (ISMRI) with a lag of 5 sesasons (MAM). Thus, the forecast skill of ISMRI were found to be higher at a lag of 5 seasons compared to a lag of 4 seasons when using the Nino-3 index. This is also found by many other studies, which the relationship between precipitation and large scale circulation signals may be stronger at longer lags (Kirono et al., 2010; Piechota et al., 1998). Thus, it is not surprise that the skill scores of the STP-BHM model are higher at longer lead times, which can also be referred as longer time lags.

[Figure]

Fig. S5. Correlation coefficient (r) between India Summer Monsoon Rainfall Index (ISMRI) and (a) Niño 1+ 2 index, (b) Niño 3 index, (c) Niño 3.4 index and (d) Niño 4 index with Niño indices lagging by 1–8 season(s) (Shukla et al., 2011).

l.414-416: Please specify what are "the BJP calibrated sub-seasonal precipitation forecasts" from Li et al. (2020). I guess it corresponds to post-processed outputs of dynamical subseasonal forecasts with a GCM, but you should remind it and give the name of the model. More generally, your assertions concerning the comparison between BHM and your previous method from Li et al (2020) should be illustrated more extensively (see Major Comment #1).

Thanks for this comment. We have compared the STP-BHM model with the NCEP model in the revised manuscript as we introduced previously.

l.436-438: "Here, we analyzed the spatial patterns of correlations between lagged signals and filtered precipitation over Region 1 at the lead time of 0-day for each step of the leave-one-year out cross-validation". I can't see where the results you are referring to are, e.g I don't know what "Here" stands for in this sentence.

Thanks for this comment. Figure S33 compares the correlation coefficient between ISO signals of U850 and precipitation for the whole period of 1979~2016 and the cross-validated period of 1980~2016 at a lead of 0-day. The results show small variability between the cross-validated correlation and the whole-period correlation. This figure has been added in the supplementary file.

[Figure]

*Figure S33. Correlation coefficient between ISO signals of U850 and precipitation for the whole period of 1979~2016 and the cross-validated period of 1980~2016.*

l.396: "The decomposition of KGE values suggest that the intraseasonal variability is underestimated in these regions" I am not sure "underestimation" is the correct word in this context. From what I understand, the important fact is that intraseasonal variability is of limited importance in those regions because it does not

account for a large fraction of total variability, so the model cannot perform well in those regions. I suggest rephrasing.

We agree with the referee that the intraseasonal variability is of limited importance in these regions. However, we mainly focus on the probabilistic forecast skill in the revised manuscript. The KGE is not used for verification any more. Thus, we have removed these sentences in the revised manuscript.

l.381-382: "The results also suggest that the probabilistic forecasts are sharp at all lead times, especially for below-normal and above normal categories". Judging by the reliability diagrams, I am not convinced by the sharpness of the forecasts. On the contrary, I think the authors should mention very limited sharpness. I guess this is intrinsic to a Bayesian approach relying on a non-informative prior.

The authors agree with the reviewer that the Bayesian approaches may have difficulty in predicting extreme events when non-informative prior is used. This indicates that the Bayesian statistical model is of limited sharpness. The copula-based statistical approaches will be used in the future to see whether the sharpness of forecasts could be improved.

**LANGUAGE AND TYPOS**

l.9: "as predictors" → "as predictor"

We have incorporated this suggestion in the revised manuscript at **L. 9**.

l. 19: "owing to the underestimation of intraseasonal variability in this region". Why underestimation?

We agree with the referee that the intraseasonal variability is of limited importance in these regions. However, we mainly focus on the probabilistic forecast skill in the revised manuscript. Thus, we have removed these sentences in the revised version.

l.22: "Other sources (…)  will be included"

We have incorporated this suggestion in the revised manuscript from **L. 26** to **L. 28** as follows:

*Other sources of sub-seasonal predictability, such as soil moisture, snow cover, and stratosphere-troposphere interaction, will be included in the future to further improve sub-seasonal precipitation forecast skill.*

l. 22: "forecast skill**s**" → "forecast skill".

I think that the word "skill" is never expected to be plural in this context. Same remark at l.34, l.74, l.116 (x2), l.395, l.425

We have incorporated this suggestion in the revised manuscript.

l.25: "mitigation**s**" → "mitigation"

We have incorporated this suggestion in the revised manuscript at **L. 31.**

l.28: "" → "launched"

The word "lunched" has been corrected as "launched" at **L. 34**.

l.30: "" → "cannot"

The word "could not" has been corrected as "cannot" at **L. 37**.

l. 32: "before it  can be used"

The word "could" has been corrected as "can" at **L. 38**.

.

l.41: "atmospheric-oceanic indices" → Do you mean "atmospheric or oceanic indices"?

The word "atmospheric-oceanic indices" has been replaced by "atmospheric or oceanic indices" at **L. 47**.

l.43: "dominant" → I suggest using another word, what about "more performant"?

The word "dominant" has been replaced by "more performant" at **L. 48**.

l.45: "plenty of"

The word "plenty" has been corrected as "plenty of" at **L. 51**.

l.48-51: "a new cluster-based empirical method (…), which the sea surface temperature (…) were included as predictors.".

The sentence is unclear, I suggest rephrasing, e.g splitting the sentence in two: "a new cluster-based method (…) European and Mediterranean regions. This method uses sea surface temperature (…) as predictors".

We have incorporated this suggestion in the revised manuscript from **L. 55** to **L. 58** as follows:

*A new cluster-based empirical method was proposed to predict winter precipitation anomalies over the European and Mediterranean Regions (Totz et al., 2017). This method used the sea surface temperature, geopotential height, sea level pressure, snow cover extent, and sea ice concentration as predictors.*

l.56: "at such a time scale" Unnecessary, please remove.

We have removed these words at **L. 63.**

l.69: "but in extra-tropical regions as well"

We have incorporated this suggestion in the revised manuscript at **L. 76**.

l.77-78: "the relationships between ISO signals and precipitation are of high uncertainty for different regions at different lead times". I suggest rephrasing, e.g "the relationships between ISO signals and precipitation are highly uncertain and depend on the region and lead time."

We have incorporated this suggestion in the revised manuscript from **L. 87** to **L. 88** as follows:

*However, we should note that the relationship between ISO signals and precipitation is highly uncertain and depend on the region and lead time.*

l.79-81: "To our best knowledge, the uncertainties of relationships between preceding ISO signals and sub-seasonal precipitation have not been fully considered in sub- seasonal precipitation forecasts in previous studies." I suggest another formulation.

We have incorporated this suggestion in the revised manuscript from **L. 91** to **L. 92** as follows:

*The uncertainty in relationship between preceding ISO signals of atmospheric field and precipitation has not been fully considered yet.*

l.84: Remove the CSC acronym. You never use it in the rest of the article.

We have removed these words at **L. 95**.

l.87: "Bayes-theorem based statistical models" → "Bayesian statistical models"

The word "Bayes-theorem based" has been replaced by "Bayesian" at **L. 98.**

l. 91: Idem

We have incorporated this suggestion in the revised manuscript at **L. 102.**

l.104: "is frequently influenced by" → "is frequently subject to"

The word "influenced by" has been replaced by "subject to" at **L. 115**.

l.111: "the model performance (…)  is evaluated"

The word "are" has been corrected as "is" at **L. 122.**

l.115-116: "the deterministic and probabilistic forecast skill is presented"

This sentence is removed as we mainly focus on probabilistic forecast skill in the revised manuscript**.**

l.127: "is area-weighted averaged over 17 hydroclimatic regions"

The word "averaging" has been corrected as "averaged" at **L. 139.**

l.134: "to monitoring" → "to monitor"

The word "monitoring" has been corrected as "monitor" at **L. 146.**

l.139: "proved to be capable of reflecting the MJO structure as the zonal wind" Unclear → "proved to be as capable of reflecting the MJO structure as the zonal wind"?

The word "as" has been added at **L. 151**.

l.148: "calculating efficiency" → "computational efficiency"?

The word "calculating" has been replaced by "computational" at **L. 160.**

l. 194, l.196: "the Africa" → "Africa"

The word "the Africa" has been corrected as "Africa" at **L. 240.**

l. 234: "in (Nardi and Rinaldo, 2011; Mcneish, 2015)". Typo, remove parentheses.

These sentences are removed as the LASSO and stepwise regression approaches are no longer used.

l.301: "A full discussion of the KGE-statistics sees Gupta et al (2009)…" → "For a full description of KGE-statistics, see Gupta et al (2009)..."

These sentences are removed as the KGE is no longer used for verification.

de Andrade, F. M., Coelho, C. A. S., and Cavalcanti, I. F. A.: Global precipitation hindcast quality assessment of the Subseasonal to Seasonal (S2S) prediction project models, Climate Dynamics, 52, 5451-5475, https://doi.org/10.1007/s00382-018-4457-z, 2019.

Hsu, P.-C., Li, T., You, L., Gao, J., and Ren, H.-L.: A spatial–temporal projection model for 10–30 day rainfall forecast in South China, Climate Dynamics, 44, 1227-1244, 10.1007/s00382-014-2215-4, 2015.

Kirono, D. G. C., Chiew, F. H. S., and Kent, D. M.: Identification of best predictors for forecasting seasonal rainfall and runoff in Australia, Hydrological Processes, 24, 1237-1247, https://doi.org/10.1002/hyp.7585, 2010.

Piechota, T. C., Chiew, F. H. S., Dracup, J. A., and McMahon, T. A.: Seasonal streamflow forecasting in eastern Australia and the El Niño–Southern Oscillation, Water Resources Research, 34, 3035-3044, https://doi.org/10.1029/98WR02406, 1998.

Shukla, R. P., Tripathi, K. C., Pandey, A. C., and Das, I. M. L.: Prediction of Indian summer monsoon rainfall using Niño indices: A neural network approach, Atmospheric Research, 102, 99-109, https://doi.org/10.1016/j.atmosres.2011.06.013, 2011.

---

## Referee Report (RR1)

**Second review of manuscript HESS-2022-67 entitled "Probabilistic sub-seasonal precipitation forecasts using preceding atmospheric intraseasonal signals in a Bayesian perspective" by Yuan Li, Zhiyong Yu, Hai He and Hao Yin**

**OVERALL RECOMMENDATION**

Minor revisions

**MAJOR COMMENTS**

The authors have satisfactorily taken into account the vast majority of my comments. I think this manuscript is far better than the previous version and can be published after correcting some typos and answering to the two points below.

1) I'm glad the authors have decided to compare the scores of their statistical forecasts to those of a dynamical S2S sytem. However, I am dubious about the choice of NCEP which is not reputed to be the best-performing one, as they acknowledge.
The authors seem to imply that except for NCEP, it is not possible "to generate pentad mean precipitation forecasts for the same period as the STP-BHM model" (l.420-421). But actually, the problem could be turned the other way round by providing STP-BHM forecasts matching the pentad mean precipitation of another S2S system, e.g ECMWF. It seems entirely feasible since STP-BHM is a purely statistical model.
I would therefore suggest a comparison with ECMWF, that would give even more importance to your statistical method if you beat it, although I consider it optional for this revision. At least, I think you should modify the sentences claiming that other comparisons are not possible (in Sections 2.1 and 3.1).

2) I am surprised to see an evaluation for both precipitation amounts and precipitation anomalies. I do not say it is irrelevant, but it is quite uncommon in the context of verification of S2S precipitation and I would have expected only one of them (presumably anomalies). I would be interested if you briefly discussed why you chose to look at both, and why the results differ. In the meantime, I think the article could be shortened by not repeating all verification charts twice (e.g the attribute diagrams and the Brier score could appear only once).

**MINOR COMMENTS**

l.119: Please specify the meaning of the STP-BHM acronym here. Although it appears in the abstract, it should be detailed once in the main body of the article.

**LANGUAGE AND TYPOS**

l.58: "which the predictors were identified" → "whose predictors were identified"

l.104: "The predictand is assumed to follow **a** distribution" (missing word)

l.124: "dataset" or "dataset**s**"?

l.149-150: "In addition, the correlation**s** between geopotential height at 850 hPa, 500 hPa, and 200 hPa (H850, H500, H200) are also analyzed." There's a missing "s" for "correlations", and the sentence is not clear. I guess you should either write "the correlations between geopotential height (…) and precipitation" or simply "the correlations with geopotential height" (as precipitation is already mentioned in the previous sentence).

l.184: "where the correlation **is** statistically significant" (missing word)

l.382 to 387, l.444 to 448: "**B**rier" (upper case B)

l. 422: "it is not **a** surprise" or "it is not **suprising**"

---

## Author Response (AR2)

**Editor decision:**

Dear authors,

One of the referees has only two minor issues for you to address. Your revised manuscript will be sent to the referee. I look forward to receiving your revised manuscript.

Sincerely,

Yi He, HESS Editor

Thanks for your kind decision. We have carefully revised the manuscript to address the comments of the reviewers. The point-by-point responses are provided in the following.

Anonymous Referee #2 reported on 26 August 2022.

Our responses are in blue and revisions are in blue and italics, with the reviewer's comments shown as normal text.

**General comments:**

The authors have satisfactorily taken into account the vast majority of my comments. I think this manuscript is far better than the previous version and can be published after correcting some typos and answering to the two points below.

Thanks for your recognition of the revised manuscript. Your constructive comments help us to greatly improve our manuscript after revision.

**Major comments:**

1) I'm glad the authors have decided to compare the scores of their statistical forecasts to those of a dynamical S2S system. However, I am dubious about the choice of NCEP which is not reputed to be the best-performing one, as they acknowledge. The authors seem to imply that except for NCEP, it is not possible "to generate pentad mean precipitation forecasts for the same period as the STP-BHM model" (l.420-421). But actually, the problem could be turned the other way round by providing STP-BHM forecasts matching the pentad mean precipitation of another S2S system, e.g ECMWF. It seems entirely feasible since STP-BHM is a purely statistical model. I would therefore suggest a comparison with ECMWF, that would give even more importance to your statistical method if you beat it, although I consider it optional for this revision. At least, I think you should modify the sentences claiming that other comparisons are not possible (in Sections 2.1 and 3.1).

Thanks for this comment. We admit that that the ECMWF model outperforms other dynamical models for S2S forecasts. It would provide stronger evidence if the STP-BHM model beats the ECMWF model. However, we also note that the forecast skill of dynamical models varies at different lead times and different regions. Figure S1 compares the forecast skill of the ECMWF model and the NCEP model from May to September over the 1999-2009 period. The ECMWF model shows higher forecast skill in **tropical regions** at all lead times. Whereas, the differences between the ECMWF model and the NCEP model are much smaller in **extratropical regions**. Both the ECMWF model and the NCEP model show little skill in predicting precipitation when the **lead time is beyond two weeks** over China. Thus, we believe that the comparison of the STP-BHM model and the NCEP model could provide useful information for sub-seasonal precipitation forecasts as well.

[Figure]

Figure S1. Correlation between the ensemble mean and observed (GPCP) accumulated precipitation anomalies for each S2S model (rows) during weeks 1-4 (columns) for hindcasts initialized from May to September over the 1999-2009 period. Correlation coefficients statistically significant at the 5% level are shaded. (Felipe et al., 2018)

To address the above comment, we have removed relevant sentences in Sections 2.1 and 3.1 as follows:
**L. 165 to L. 168:** The STP-BHM model we built in this study is compared to the dynamical models to provide a benchmark for sub-seasonal precipitation forecasts. In this study, we compare our results of the STP-BHM model with the NCEP model archived in the S2S Database for the same period of 1999-2010 from May to October (http://apps.ecmwf.int/datasets/data/s2s/).

**L. 416 to L. 421:** Figure 9 compares the CRPS skill scores of the STP-BHM model and the NCEP model from May to October during the period of 1999~2010. It is not surprising that the NCEP model outperforms the STP-BHM model when the lead time is within 5 days. However, we should note that the STP-BHM model shows much higher probabilistic forecast skill compared to the NCEP model at longer lead times. Positive CRPS skill scores are observed for the STP-BHM model over most hydroclimatic regions, whereas the skill scores are mostly negative for the NCEP model.

2) I am surprised to see an evaluation for both precipitation amounts and precipitation anomalies. I do not say it is irrelevant, but it is quite uncommon in the context of verification of S2S precipitation and I would have expected only one of them (presumably anomalies). I would be interested if you briefly discussed why you chose to look at both, and why the results differ. In the meantime, I think the article could be shortened by not repeating all verification charts twice (e.g the attribute diagrams and the Brier score could appear only once).
Thanks for this comment. In our first submission, we built the statistical model for pentad mean precipitation amounts as it could be further used for hydrological predictions at sub-seasonal time scales. The anonymous referee #1 suggested that we should verify the prediction skill for precipitation anomalies as well. To have a broader audience, we finally chose to predict both precipitation amounts and precipitation anomalies.

We added a brief explanation of the results in the discussion section from **L. 480** to **L. 485** as follows:
We also note that the forecast skill of pentad mean precipitation amounts and precipitation anomalies are different. The precipitation amounts behavior at different timescales (interannual, intraseasonal, and synoptic). The large-scale circulation anomalies (U850, U200, H850, and H500) may be dominant for the total variability of precipitation amounts. In comparison, the precipitation anomalies only represent the intraseasonal component of precipitation. The OLR plays a more important role for intraseasonal convections compared to other dynamical fields (Ventrice et al., 2013; Liu et al., 2016).

To shorten the length of the manuscript, we have moved Fig. 11 and Fig. 13 to the supplementary file.

Liu, P., Zhang, Q., Zhang, C., et al. A revised real-time multivariate MJO index. *Monthly Weather Review*, 144(2), 627-642, 2016.
Ventrice, M. J., Wheeler, M. C., Hendon, H. H., et al. A Modified Multivariate Madden–Julian Oscillation Index Using Velocity Potential, *Monthly Weather Review*, 141(12), 4197-4210, 2013.

**MINOR COMMENTS**

l.119: Please specify the meaning of the STP-BHM acronym here. Although it appears in the abstract, it should be detailed once in the main body of the article.

Thanks for this comment. We have incorporated this suggestion in the revised manuscript at **L. 119**.

**LANGUAGE AND TYPOS**

l.58: "which the predictors were identified" → "whose predictors were identified"

The word "which" has been corrected as "whose" at **L. 58**.

l.104: "The predictand is assumed to follow **a** distribution" (missing word)

The word "a" has been added at **L. 104**.

l.124: "dataset" or "dataset**s**"?

The word "dataset" has been corrected as "datasets" at **L. 125**.

l.149-150: "In addition, the correlations between geopotential height at 850 hPa, 500 hPa, and 200 hPa (H850, H500, H200) are also analyzed." There's a missing "s" for "correlations", and the sentence is not clear. I guess you should either write "the correlations between geopotential height (…) and precipitation" or simply "the correlations with geopotential height" (as precipitation is already mentioned in the previous sentence).

This sentence has been rewritten from **L. 150** to **L. 151** as follows:

In addition, the correlations with geopotential height at 850 hPa, 500 hPa, and 200 hPa (H850, H500, H200) are also analyzed.

l.184: "where the correlation **is** statistically significant" (missing word)

The word "is" has been added at **L. 182**.

l.382 to 387, l.444 to 448: "**B**rier" (upper case B)

We have incorporated this suggestion in the revised manuscript.

l. 422: "it is not a surprise" or "it is not **surprising**"

We have incorporated this suggestion in the revised manuscript at **L. 417**.